# $N_2$ cleavage by silylene and formation of $H_2Si(\mu\text{-N})_2SiH_2$

Liyan Cai[1], Bing Xu [ID][1] ✉, Juanjuan Cheng[1], Fei Cong[1], Sebastian Riedel [ID][2] ✉ & Xuefeng Wang [ID][1] ✉

Fixation and functionalisation of $N_2$ by main-group elements has remained scarce. Herein, we report a fixation and cleavage of the N ≡ N triple bond achieved in a dinitrogen ($N_2$) matrix by the reaction of hydrogen and laser-ablated silicon atoms. The four-membered heterocycle $H_2Si(\mu\text{-N})_2SiH_2$, the $H_2SiNN(H_2)$ and HNSiNH complexes are characterized by infrared spectroscopy in conjunction with quantum-chemical calculations. The synergistic interaction of the two $SiH_2$ moieties with $N_2$ results in the formation of final product $H_2Si(\mu\text{-N})_2SiH_2$, and theoretical calculations reveal the donation of electron density of Si to π* antibonding orbitals and the removal of electron density from the π bonding orbitals of $N_2$, leading to cleave the non-polar and strong NN triple bond.

The industrial synthesis of $NH_3$ relies on the transition metal-catalyzed Haber-Bosch process[1–3], in which the inert dinitrogen is converted to ammonia under harsh reaction conditions. This N≡N triple bond activation is based on partially filled *d*-orbitals of the d-block elements (e.g., Fe), which have a suitable symmetry and energy. An alternative way to activate dinitrogen is through main-group element compounds via the π back-donation from either the *d* orbitals (Ca, Sr, Ba) or *p* orbitals (Be, B, C)[4]. Wherein *p*-block elements (e.g., B and C) have a non-bonding (donor) electron pair and an energetically low-lying vacant (acceptor) orbital which can mimic the *d*-orbital character to activate dinitrogen[5–7]. Recently the *p*-block element boron was successfully used to activate dinitrogen[8–10]. Five- and seven-electron boron-centered radicals ($R_2B\cdot$) are predicted to be favorable for dinitrogen activation both thermodynamically and kinetically[11,12]. A crucial metric for nitrogen activation of substantially elongated N–N bond has been achieved by a borylene coordinating an $N_2$ molecule in an end-on bridging position as [{(CAAC)-DurB}$_2$($\mu^2$-$N_2$)] (CAAC = cyclic alkylamino carbene, Dur = 2,3,5,6-tetramethylphenyl)[13–15]. Our groups reported a complete cleavage of the N≡N triple bond by fluoroborylene (:BF) which has been observed as a cyclic FB($\mu$-N)$_2$BF system in a matrix-isolation investigation under cryogenic conditions[16]. Carbene, another reactive intermediate, has also been used for $N_2$ activation and conversion[17–19]. Maier et al. found that singlet $\sigma^0\pi^2$ carbene (2-diazo-2H-imidazole) would bind with dinitrogen in the matrix,

demonstrating the potential for $\sigma^0\pi^2$ carbene to activate dinitrogen[20]. Furthermore, $N_2$ activation by a carbene pair has been calculated and the N≡N triple bond was predicted to be elongated to an N-N single bond (1.428 Å) under the synergistic effect of the two $CH_2$ moieties[21]. Silylene, the silicon analog of carbene, has been proven to exhibit similar properties to transition metal compounds, as it has a narrow HOMO-LUMO energy gap, which has received particular attention for the activation of small molecules[22]. As reported in 1998, the first isolable N-heterocyclic silylene reacts rapidly with dry $O_2$, giving rise to a colorless and insoluble disiladioxetane polymer[23]. From then on, many small stable molecules, such as $CO_2$[24], $H_2O$[25], $P_4$[26], $C_2H_4$[27], $H_2$[28], $NH_3$[29], and C-H bonds[30] have been activated by silylenes (see Fig. 1). In 2019 the splitting and reductive homocoupling of CO was achieved using the bis-silylenes $(LSi:)_2$Xant [Xant = 9,9-dimethylxanthene-4,5-diyl; L = PhC(NtBu)$_2$] and $(LSi:)_2$Fc (Fc = 1,10-ferrocenyl)[31]. So far, only homoleptic $N_2$ complexes of silicon have been reported under matrix-isolation conditions, including SiNN, Si(NN)$_2$[32], and larger silicon-nitrogen clusters[33,34], but the activation of more inert nitrogen by silylenes remain a challenge, although silylenes do exhibit high reactivity for other small molecules[24–31]. The key difficulty in silylene-mediated nitrogen activation is to modify the occupied and vacant orbitals of silylene in space and energy, which could enhance the weakening and functionalization of an inert chemical bond. For example, Driess et al. reported that two silylene moieties (bis-silylenes)

[1]School of Chemical Science and Engineering, Shanghai Key Lab of Chemical Assessment and Sustainability, Tongji University, Shanghai 200092, China. [2]Institut für Chemie und Biochemie – Anorganische Chemie, Freie Universität Berlin, Fabeckstrasse 34-36, D-14195 Berlin, Germany. ✉e-mail: xbrare@tongji.edu.cn; s.riedel@fu-berlin.de; xfwang@tongji.edu.cn

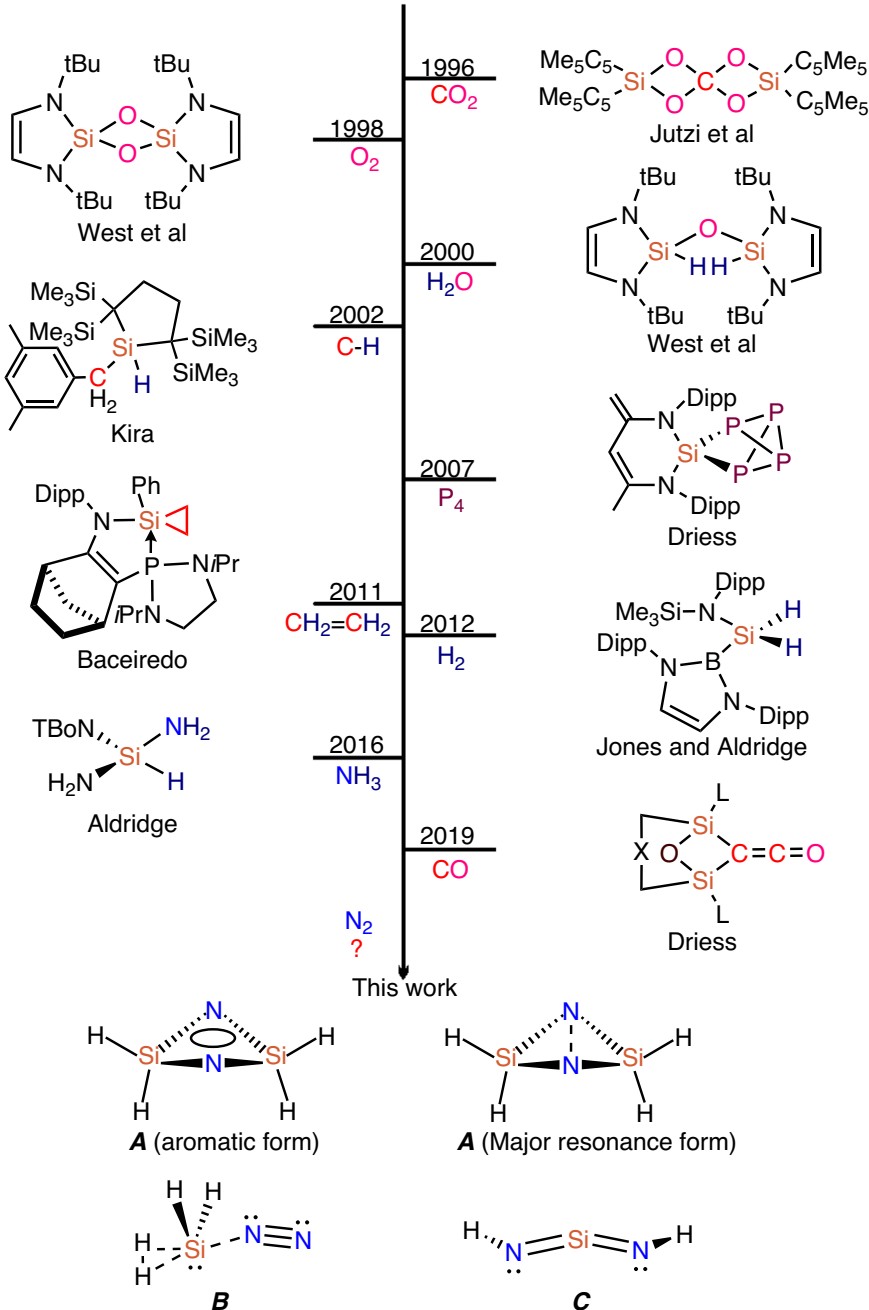

**Fig. 1 | Binding modes of small molecules to silylene.** (Dipp = 2,6-$^i$Pr$_2$C$_6$H$_3$; TBoN = N(SiMe$_3$){B(DippNCH)$_2$}; L = PhC(N$t$Bu)$_2$; X = 9, 9-dimethyl-xanthene- 4, 5-diyl or 1, 1′ - ferrocenyl).

could be cooperative in cleaving unreactive bonds, in which the Si---Si distance plays a crucial role[31,35].

Here we report on the activation of dinitrogen by silylenes under cryogenic conditions. In our matrix-isolation experiments, laser-ablated silicon atoms have been reacted with H$_2$, D$_2$, HD, and H$_2$/D$_2$ mixtures in solid nitrogen that served as both reactant and host-matrix material. H$_2$Si($\mu$-N)$_2$SiH$_2$ (**A**) is spectroscopically and quantum-chemically characterized, which demonstrates that the N$_2$ triple bond is cleaved by the synergistic interaction of the two SiH$_2$ moieties. In addition H$_2$SiNN(H$_2$) (**B**) and HNSiNH (**C**) are identified.

## Results

Figures 2 and 3 show infrared spectra obtained after the laser-ablation of Si atoms co-deposited with a 10 % H$_2$/N$_2$ mixture at 4 K in

a dinitrogen matrix. Further details are provided in the Supplementary Information (Supplementary Figs. 1–6). Isotopic experiments with H$_2$, D$_2$, HD, and H$_2$/D$_2$ samples in pure $^{14}$N$_2$, $^{15}$N$_2$, and $^{14/15}$N$_2$ mixtures together with frequency calculations at the DFT level were used for the product identification (see Table 1 and Supplementary Tables 1–6). The EDA-NOCV method was used to elucidate the peculiar stability of the bonding nature. In addition to the three adduct products **A**, **B**, and **C**, H$_2$SiNN, Si(NN)$_2$, and SiNN have been observed (Fig. 1 and Supplementary Tables 1, 7)[32,36,37]. The absorptions associated with three silicon-nitrogen hydrides H$_2$Si($\mu$-N)$_2$SiH$_2$ (**A**), H$_2$SiNN(H$_2$) (**B**), and HNSiNH (**C**) were unambiguously assigned based on their growth/decay behavior in different experiments and on their characteristic H/D and $^{14/15}$N isotope pattern. The absorptions for silicon nitrides and hydrides such as SiNN, Si(NN)$_2$, SiH$_2$, and

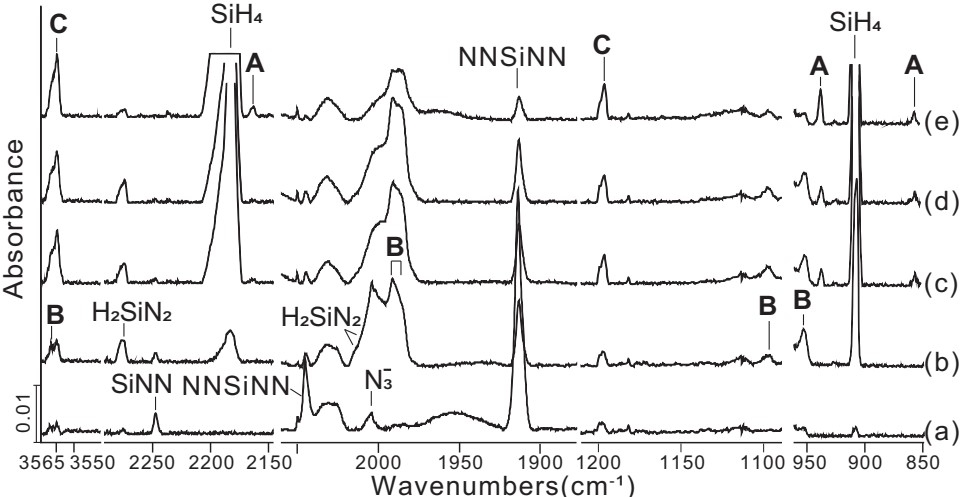

**Fig. 2 | Infrared spectra of the reaction products of laser-ablated Si atoms with hydrogen under an excess of solid nitrogen at 4 K. a** Codeposition of Si + 10% H₂ for 120 min; **b** after λ > 300 nm irradiation for 10 min; **c** after λ > 220 nm irradiation for 10 min; **d** after annealing to 7 K; **e** after λ > 220 nm irradiation for 10 min. A H₂Si(μ-N)₂SiH₂, B H₂SiNN(H₂), C HNSiNH.

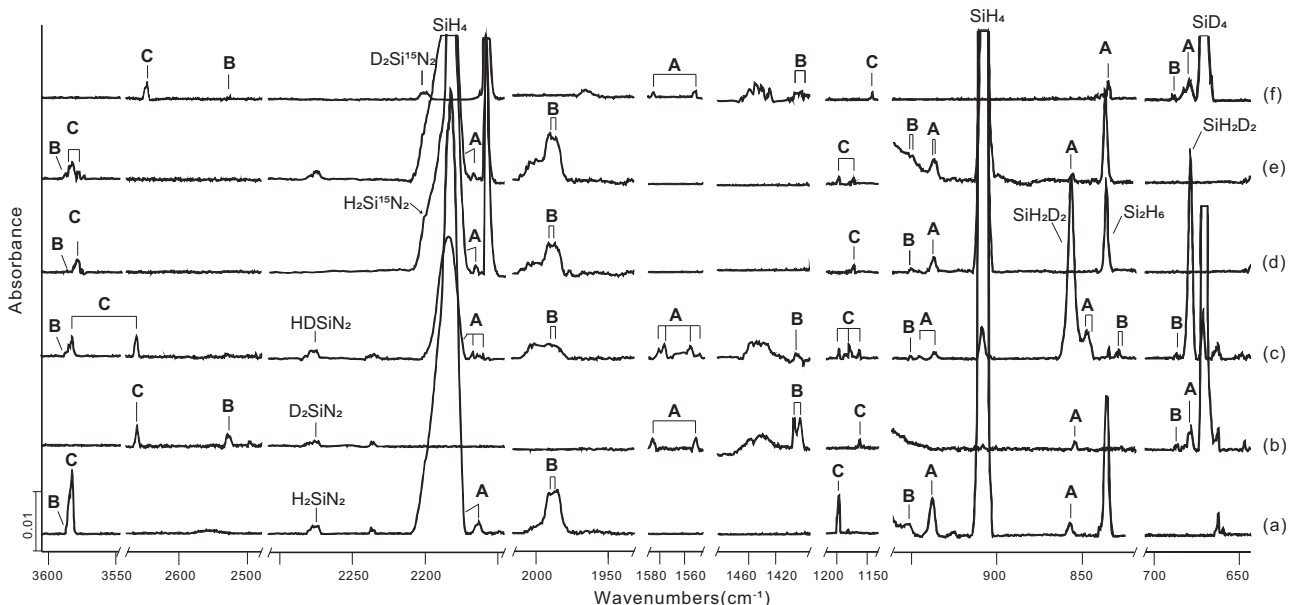

**Fig. 3 | Infrared spectra of the laser-ablated Si atoms reactions with H₂, D₂ and HD in excess solid ¹⁴N₂ (¹⁵N₂) at 4 K after the second λ > 220 nm irradiation for 10 min. a** Si + 10% H₂/¹⁴N¹⁴N; (**b**) Si + 10% D₂/¹⁴N¹⁴N; **c** Si + 15% HD/¹⁴N¹⁴N; **d** Si + 10% H₂/¹⁵N¹⁵N; **e** Si + 10% H₂/¹⁴N¹⁴N + ¹⁵N¹⁵N; **f** Si + 10% D₂/¹⁵N¹⁵N. A H₂Si(μ-N)₂SiH₂, B H₂SiNN(H₂), C HNSiNH.

SiH₄ have been reported previously and agree with our assignments (Supplementary Figs. 7, 8)[32,38,39].

The absorptions of the cyclic species **A** were observed upon λ > 220 nm irradiation and increased markedly upon the second λ > 220 nm irradiation (Fig. 2). The symmetric and antisymmetric D-Si-D modes were observed at 1551.3 and 1586.1 cm⁻¹, respectively (Fig. 3b and Supplementary Fig. 1). Only the symmetric H-Si-H mode at 2165.3 cm⁻¹ can be detected, leading to a H/D ratio of 1.3958. This observation is most likely due to a strong SiH₄ band that covers the antisymmetric mode. In addition, the strongest absorption at 937.8 cm⁻¹ of the H-Si-H bending mode shifted to 679.2 cm⁻¹ in the deuterium experiment (H/D isotopic ratio 1.3807) (Fig. 3b and Supplementary Fig. 1), and in the ¹⁵N₂ experiment a slight shift of 0.6 to 937.2 cm⁻¹ was observed (Fig. 3d and Supplementary Fig. 4).

Furthermore, a band for species **A** at 855.3 cm⁻¹ in the D₂/¹⁴N₂ experiment (Fig. 3b) shifted by 20.6 to 834.7 cm⁻¹ in the D₂/¹⁵N₂ experiment (Fig. 3f and Supplementary Fig. 6), which results in a ¹⁴N/¹⁵N ratio of 1.0247. The band can be assigned to the Si-N-Si stretching mode. The corresponding H₂/¹⁴N₂ experiment shows a band at 857.7 cm⁻¹ (Fig. 3a). Its counterpart in the H₂/¹⁵N₂ experiment was covered by a broad signal of Si₂H₆ centered at 835.4 cm⁻¹ (Fig. 3d and Supplementary Fig. 4).

When the isotopologues HD or a mixture of 5% H₂ and 5% D₂ as a hydrogen source were used, band distributions were observed for the bands of species **A** (Fig. 3c and Supplementary Figs. 2, 3). There are four possible isomers of **A** (**A₃, A₄, A₅**, and **A₆**) relating to 2H and 2D (see Supplementary Tables 1, 2). The 15% HD/¹⁴N₂ experiment shows bands at 2167.4 and 2162.4 cm⁻¹ (Si-H symmetric vibration), 1580.1, 1575.8, 1556.8, and 1550.5 cm⁻¹ (Si-D vibration), 847.5 and 845.3 cm⁻¹

**Table 1 | Observed and calculated (CCSD(T)/aug-cc-pVDZ) infrared absorptions (cm⁻¹) and H/D ratios for products of the reactions of Si atoms with $H_2$ molecules in solid $N_2$**

| | $H_2$ | | $D_2$ | | H/D ratio | | Mode assignment |
|---|---|---|---|---|---|---|---|
| | Calc. | Obs. | Calc. | Obs. | Calc. | Obs. | |
| **A** $H_2Si(\mu$-N$)_2SiH_2$ ($^1A_1$, $C_{2v}$) | | | | | | | |
| $^{14}$N | 2275.3 (150) | covered | 1642.0 (76) | 1586.1 | 1.3857 | / | $H_2$Si-Si$H_2$ stretch, $b_2$ |
| | 2230.9 (163) | 2165.3 | 1601.7 (69) | 1551.3 | 1.3928 | 1.3958 | $H_2$Si-Si$H_2$ stretch, $a_1$ |
| | 964.4 (308) | 937.8 | 689.5 (79) | 679.2 | 1.3988 | 1.3807 | 2Si$H_2$ bend, $b_2$ |
| | 874.3 (170) | 857.7 | 872.6 (288) | 855.3 | 1.0019 | 1.0028 | Si(NN)Si ring, $b_2$ |
| $^{15}$N | 2275.2 (137) | covered | 1642.0 (75) | 1586.1 | 1.3856 | / | $H_2$Si-Si$H_2$ stretch, $b_2$ |
| | 2230.8 (161) | 2165.3 | 1601.7 (68) | 1551.3 | 1.3928 | 1.3958 | $H_2$Si-Si$H_2$ stretch, $a_1$ |
| | 964.4 (287) | 937.2 | 688.5 (75) | 679.2 | 1.4008 | 1.3799 | 2Si$H_2$ bend, $b_2$ |
| | 855.7 (180) | covered | 854.9 (282) | 834.7 | 1.0009 | / | Si(NN)Si ring, $b_2$ |
| **B** $H_2$SiNN($H_2$) ($^1A_1$, $C_s$) | | | | | | | |
| $^{14}$N | 3733.3 (86) | 3569.3 | 2641.2 (43) | 2530.6 | 1.4135 | 1.4105 | H-H stretch, a' |
| | 2034.4 (215) | 1991.6 | 1464.3 (110) | 1425.9 | 1.3894 | 1.3967 | Si$H_2$ stretch, a'' |
| | 2024.8 (143) | 1983.6 | 1454.4 (76) | 1421.8 | 1.3922 | 1.3951 | Si$H_2$ stretch, a' |
| | 1129.3 (50) | 1097.5 | 803.8 (26) | 804.4 | 1.4050 | 1.3644 | H-H twist, a' |
| | 982.2 (78) | 952.4 | 704.5 (39) | 684.4 | 1.3942 | 1.3916 | Si$H_2$ scissor, a' |
| $^{15}$N | 3733.3 (86) | 3569.3 | 2641.2 (43) | 2530.6 | 1.4135 | 1.4105 | H-H stretch, a' |
| | 2034.3 (215) | 1991.6 | 1464.3 (111) | 1425.9 | 1.3893 | 1.3967 | Si$H_2$ stretch, a'' |
| | 2024.7 (143) | 1983.6 | 1454.4 (76) | 1421.8 | 1.3921 | 1.3951 | Si$H_2$ stretch, a' |
| | 1129.2 (50) | 1097.5 | 803.8 (26) | 804.4 | 1.4050 | 1.3644 | H-H twist, a' |
| | 982.2 (78) | 952.4 | 704.5 (39) | 684.4 | 1.3941 | 1.3916 | Si$H_2$ scissor, a' |
| **C** HNSiNH ($^1A_1$, $C_2$) | | | | | | | |
| $^{14}$N | 3559.9 (176) | 3564.8 | 2607.4 (124) | 2662.5 | 1.3653 | 1.3389 | NH stretch, $u_{as}$ |
| | 1228.1 (106) | 1197.6 | 1194.2 (128) | 1164.8 | 1.0284 | 1.0282 | NSiN stretch, $u_{as}$ |
| $^{15}$N | 3551.8 (132) | 3556.2 | 2595.2 (120) | 2647.3 | 1.3686 | 1.3433 | NH stretch, $u_{as}$ |
| | 1211.9 (37) | 1175.9 | 1177.8 (123) | 1143.2 | 1.0290 | 1.0286 | NSiN stretch, $u_{as}$ |

(SiHD bend) which correspond to the isomers $A_4$ (2167.4, 1575.8, and 847.5 cm⁻¹), $A_5$ (1556.8 and 845.3 cm⁻¹) and $A_6$ (1580.1, 1550.5, and 845.3 cm⁻¹). The experiments using a mixture of 5% $H_2$/5% $D_2$ in $^{14}N_2$ show three main isomers $A_1$, $A_2$, $A_3$ with four groups of double bands at 2165.3 and 2163.0 cm⁻¹ (Si-H vibration), 1586.1, 1577.2, 1551.3, and 1549.2 cm⁻¹ (Si-D vibration), 946.4 and 937.8 cm⁻¹ (Si$H_2$ and SiD$_2$ bend), 857.7 and 855.3 cm⁻¹(Si(NN)Si ring); among these the bands at 2163.0, 1577.2, 1549.2, and 946.4 cm⁻¹ can be attributed to $A_3$. All bands are listed in Table 1 and Supplementary Tables 1–3. The identification of molecule **A** is also based on the excellent agreement of the observed and calculated frequencies at the CCSD(T) and B3LYP level of theory.

The calculated N–N distance in **A** is 1.828 Å at the B3LYP level (Fig. 4), which is significantly longer than the N−N single bond of diphenylhydrazine [d(N-N): 1.394 Å][40]. A Mayer bond order of the N−N bond for molecule **A** is 0.689 computed at the B3LYP/6-311 G(3$df$,3$pd$) level. This suggests that the N≡N triple bond is cleaved by the two Si$H_2$ units. The resonance structures of compound **A** have been provided by NBO-based Natural Resonance Theory (NRT) analysis (Supplementary Fig. 9)[41–43]. The ring inversion barrier of the puckered ring system of **A** was computed to be 6.2 kcal mol⁻¹ at the B3LYP/6-311 + + G (3df, 3pd) level (Supplementary Fig. 10). As known from other cyclic main-group species, aromaticity is an important factor to stabilize both the transition state and the product during the $N_2$ fixation process. To evaluate the compound's aromaticity, computational chemistry is an effective tool[44,45], and the aromaticity of the four-membered ring in **A** has been confirmed by the multi-center bond order (MCBO) indexes, gauge including magnetically induced current (GIMIC)[46,47], the nucleus-independent chemical shift (NICS)[48] and the electron density of delocalized bonds (EDDB)[49,50] analyses. MCBO index, which is also known as the multi-center index (MCI) to evaluate aromaticity from the aspect

of electron delocalization properties[51–53]. The MCBO index of **A** is 0.3046, similar to that of FB($\mu$-N$)_2$BF (0.3190), which also has an aromatic four-membered $B_2N_2$ ring[16]. GIMIC method was calculated at B3LYP/6-311 + +g (3$df$, 3$pd$) level and a net diamagnetic ring current of 11.4 nA T⁻¹ in **A** similar to the typical aromatic molecule benzene (11.8 nA T⁻¹)[54] could demonstrate the aromaticity of complex **A** (Supplementary Fig. 11). What's more, the isotropic magnetic shielding tensor has been the most popular index for measuring aromaticity[55], and the average NICS(1)$_{av}$ index can serve as a probe of aromaticity in nonplanar molecular systems. (Supplementary Fig. 12)[56]. The large negative NICS(1)$_{av}$ and NICS(0) indexes of −29.0 and −39.1 obtained at the center of gravity of species **A** indicate its aromatic character. Canonical molecular orbital natural chemical shielding (CMO-NICS(1)$_{ZZ}$)[57] was calculated at B3LYP/ 6-311 + + G(3$df$,3$pd$) level to separate the σ and π contributions of canonical molecular orbital, and larger diatropic contribution of −14.6 ppm from σ orbitals compared with −11.9 ppm from π orbitals (Supplementary Fig. 13) indicates σ aromaticity dominated in **A**, consistent with the EDDB analysis where 1.55 electrons delocalization was calculated (Supplementary Table 8) which are comparable to the number of delocalized electrons in $L_2Si_2P_2$ (L = PhC(NtBu)$_2$, 1.77 electrons)[58] and the tetraatomic boron specie (1.57 electrons)[59] and σ aromaticity is also the dominant one (60%).

$H_2$SiNN($H_2$) (**B**) shows absorptions at 3569.3, 1991.6, 1983.6, 1097.5, and 952.4 cm⁻¹ which appeared already upon codeposition (Fig. 2) and markedly increased upon λ > 300 nm irradiation with the obvious reduction of the SiNN. It decreased by 60% upon λ > 220 nm irradiation. The strong bands at 1991.6, 1983.6, and 952.4 cm⁻¹ are assigned to the Si-H stretching and bending modes of the Si$H_2$ subunit, which shifted to 1425.9, 1421.8, 684.4 cm⁻¹ in the deuterium

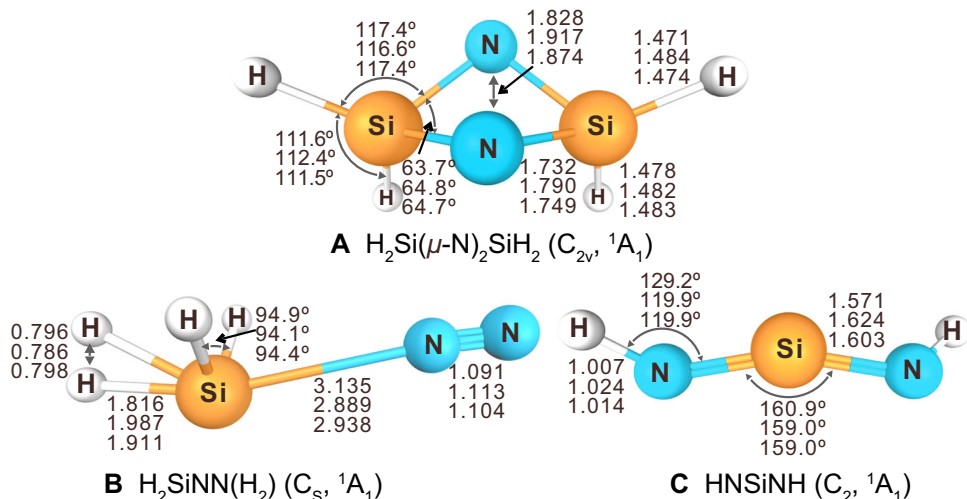

**Fig. 4 | Calculated geometries of A, B, and C.** $H_2Si(\mu\text{-}N)_2SiH_2$ (**A**), $H_2SiNN(H_2)$ (**B**), HNSiNH (**C**) based on B3LYP/6-311 + + G(3*df*,3*pd*) (upper), CCSD(T)/aug-cc-pVDZ (middle), and CCSD(T)/aug-cc-pVTZ (lower) methods (bond lengths in Å and bond angles in degree).

experiment (Fig. 3b and Supplementary Fig. 1), giving isotopic frequency ratios of 1.3967, 1.3951, and 1.3916, respectively. Moreover, two weak bands located at 3569.3 and 1097.5 cm$^{-1}$ have been observed with $H_2$, which shift to 2530.6 and 804.4 cm$^{-1}$ with $D_2$, exhibiting a H/D ratio of 1.4105 and 1.3644. They can be assigned to an H-H stretching and bending mode (Supplementary Fig. 1). These bands appear at 3110.3, 3102.4, and 977.0 cm$^{-1}$ in the HD sample (Supplementary Fig. 2). Some bands at 1987.2, 1984.5, 1425.3, and 830.0 cm$^{-1}$ turn up in 5% $H_2$/5% $D_2$ or HD experiments (Fig. 3c and Supplementary Figs. 2, 3) and are assigned to a SiHD stretching and bending mode. All these results suggest that **B** contains one side-on coordinated $H_2$ molecule. The computed frequencies of $H_2SiNN(H_2)$ are provided in Table 1 and Supplementary Table 4 and show a very good agreement between theory and experiment. Furthermore, in low $H_2$ concentrations (less than 2%), sharp bands at 2274.2, 2013.6, 2009.4, and 928.8 cm$^{-1}$ were observed, which have been assigned to $H_2SiN_2$ (Supplementary Figs. 14, 15)[36]. As shown in Supplementary Fig. 16, with an increase of $H_2$, these doublet bands disappear, and the absorptions of species **B** will strongly be enhanced. This suggests an additional $H_2$ coordination to $H_2SiN_2$ and, therefore, built species **B**. The ν(Si-H) stretching for **B** at 1991.6 and 1983.6 cm$^{-1}$ is lower than these modes for $H_2SiN_2$ and the ν(N-N) stretching is higher, which is in good agreement with the B3LYP and CCSD(T) calculations (Supplementary Tables 9, 10). We performed Tesla coil discharge experiments to confirm the assignment of compound **B** (Supplementary Figs. 17–19). The absorptions of compound **B** increased after annealing to 33 K at the expense of $H_2SiN_2$ (Supplementary Fig. 17) and became even more intense with the increase of $H_2$ concentrations ranging from 0 to 10% (Supplementary Fig. 18). This proves that $H_2$ plays an important role in coordinating to $H_2SiN_2$ to give species **B**. Compound **B** might be described as a pseudo $SiH_4$ forming a very weak adduct with dinitrogen with a bond distance slightly shorter than the van der Waals distance of 3.7 Å and the N-N stretching frequency is too weak to be observed.

Two additional bands at 3564.8 and 1197.6 cm$^{-1}$ assigned to a new molecule HNSiNH (**C**) have been observed by codeposition. They increased on both λ > 300 nm and λ > 220 nm irradiation at the expense of the absorptions of $H_2SiN_2$ and SiNN (see Fig. 2 and Supplementary Tables 5, 11). These bands belong to the N–H and N–Si–N stretching modes of the species HNSiNH. Additional isotope experiments were performed using a mixture of $D_2/N_2$. They show absorptions of the corresponding isotopologues at 2662.5 cm$^{-1}$ (H/D isotopic ratio 1.3389), which is very close to the isotopic ratio of HNSi in a 4 K argon matrix (H/D isotopic ratio 1.3424)[60], and at 1164.8 cm$^{-1}$ (H/D

isotopic ratio 1.0282), see Fig. 3b and Supplementary Fig. 1. With $H_2$/$^{15}N_2$, the counterpart bands were observed at 3556.2 cm$^{-1}$ ($^{14}N$/$^{15}N$ isotopic ratio 1.0024) and 1175.9 cm$^{-1}$ ($^{14}N$/$^{15}N$ isotopic ratio 1.0186) also shown in Fig. 3d and Supplementary Fig. 4. With $D_2$/$^{15}N_2$, the two bands shifted to 2647.3 and 1143.2 cm$^{-1}$ (Fig. 3f and Supplementary Fig. 6). Moreover, in our HD experiment, a band at 1178.4 cm$^{-1}$ reveals an isotopic triplet indicating the involvement of two equivalent hydrogen atoms (Fig. 3c and Supplementary Figs. 2, 3). The Si–N bond length of **C**, calculated to be 1.571 Å at the B3LYP level (Fig. 4), is much shorter than the Si–N single bond (1.87 Å) and is close to the N=Si bond in HN=Si (1.559 Å)[61–64] and the N=Si monomer with a double bond of 1.572 Å[65,66].

As shown in Fig. 5, SiNN ($^3\Sigma$) was supposed to be a starting compound to give the complex $H_2SiNN$ ($^1A_1$) with $H_2$ upon λ > 300 nm irradiation. In the next step, $H_2SiNN$ reacts with a second $H_2$ molecule to form complex **B**. The corresponding barrier was computed to be only 1.8 kcal/mol at the DFT level, which is in accordance with the increase of $H_2SiNN$ and complex **B** and the decrease of SiNN upon λ > 300 nm irradiation. Although $H_2Si$: could not be observed directly in the spectrum, it can still be assumed that $H_2Si$: is formed either in the reaction of silicon with hydrogen or via $SiH_x$ decomposition, and in a dinitrogen atmosphere will further react with $N_2$ or $N_2$ and $H_2$ to form $H_2SiNN$ and $H_2SiNN(H_2)$. Further reaction of $H_2SiNN(H_2)$ with SiNN forms the $H_2Si(\mu\text{-}N)_2SiH_2$ (**A**) complex through hydrogen transfer, forming two $SiH_2$ subunits and finally leading to an NN triple bond cleavage (Fig. 5 and Supplementary Figs. 20–22). In principle, the cryogenic conditions of the matrix provide a good environment in that both $H_2SiNN(H_2)$ and SiNN synergistically react as Lewis bases and initiate a step-wise reduction of the $N_2$ moiety. In the Tesla coil discharge reactions of $SiH_4$ with or without $H_2$ in excess solid $N_2$, both $H_2SiN_2$ and species **B** can be observed, while species **A** is missing due to the lack of SiNN (Supplementary Fig. 23g–i). In the reactions of laser-ablated Si atoms with 10% $SiH_4$ in solid nitrogen, species **A** is also not generated because of the absence of species **B**. This shows that both **B** and SiNN are essential for the formation of product **A** (Supplementary Fig. 23d–f).

Based on quantum-chemical calculations, the nitrogen atoms in **A** show a more negative partial charge of −1.07 e, while the Si atoms carry a more positive partial charge of 1.45 e compared to compound **B** (Supplementary Table 12). As shown at the energy surface (Fig. 5), A is computed to be exothermic by 114.6 kcal mol$^{-1}$ with the highest activation barrier of 38.4 kcal mol$^{-1}$. Note that the activation of CO to react to the ethynediolate dianion [OCCO]$^{2-}$ has been achieved by a bissilylene[31]. In addition, SiNN ($^1A_1$, $C_{2v}$) also leads to the formation of **C**,

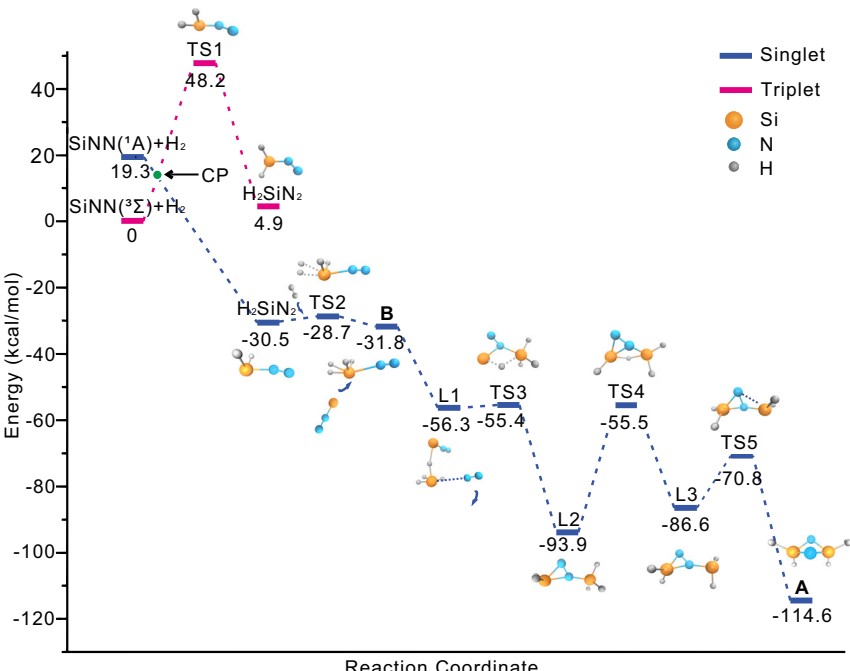

**Fig. 5 | Potential energy surface for the reaction of *B* + SiNN → *A* + N₂ computed at the B3LYP/6-311 + + G(3*df*, 3*pd*) level of theory.** The unit of relative energy is kcal/mol.

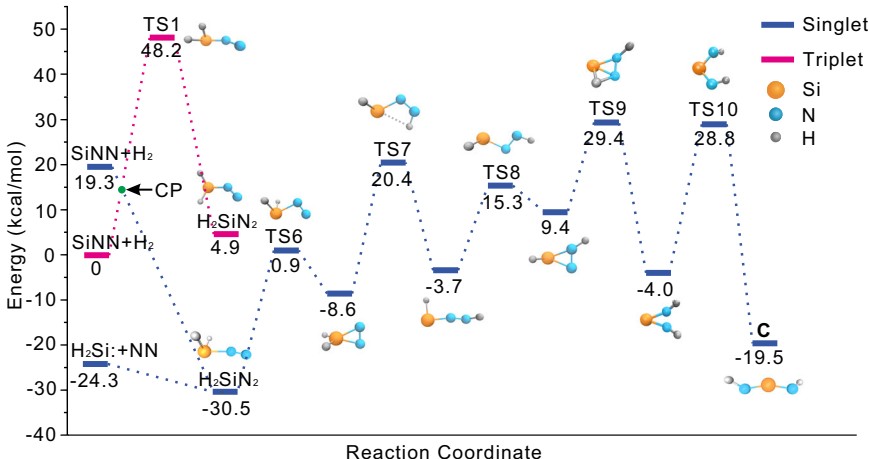

**Fig. 6 | Potential energy surface for the reaction of H₂Si + N₂ → *C* and SiNN + H₂ → *C* computed at the B3LYP/6-311 + + G(3*df*, 3*pd*) level of theory.** The unit of relative energy is kcal/mol.

with an activation barrier of 32.8 kcal mol⁻¹. **C** is observed in freshly deposited samples, and it increased markedly after λ > 220 nm irradiation, in which the laser-ablated energy provided in the codeposition process and the irradiation energy at λ > 220 nm most likely support the formation separately. SiNN (¹A₁, C₂ᵥ) reacts with H₂ to form H₂SiN₂ and then isomerizes to **C** through H transfer and N-N bond cleavage (Fig. 6 and Supplementary Figs. 24, 25). Similar exothermic reactions could occur when H-substituted groups, such as CH₃ and Ph, are applied (Supplementary Fig. 26). Similar to **NHC1-H**[67], H₂Si displays the smallest singlet–triplet energy gap and lowest ΔE to give H₂SiNN, showing the great potential for dinitrogen activation.

The EDA-NOCV method was used to elucidate the peculiar stability of the bonding nature in **A**. As shown in Fig. 7 the neutral fragments N₂ (¹Σg) and (SiH₂)₂ (¹A) in the singlet state, which refer to the symmetry-

allowed dissociation products, have been selected as interacting moieties that address the question about all changes along the bond formation between two neutral fragments[68]. The numerical results are shown in Supplementary Table 13 and the breakdown of the orbital interaction into pairwise orbital interactions reveals that the dominant orbital stabilization, $\Delta E_{orb(1)}$ (−329.1 kcal mol⁻¹) and $\Delta E_{orb(2)}$ (−318.7 kcal mol⁻¹) comes from the back-donation of the HOMO-1 (mainly 3$p_x$ of the Si atom) and the HOMO (mainly 3$p_x$ of the Si atom) of the (SiH₂)₂ moiety into the two perpendicular π* MOs of the N₂ ligands, known as push effect. The most interesting orbital interactions are $\Delta E_{orb(3)}$ and $\Delta E_{orb(4)}$, which contribute to the donation of π MO electrons of the N₂ ligands to the LUMO + 1 (3$p_z$ of Si atom and 1$s$ of H atom) and LUMO (mainly 3$s$, 3$p_y$, 3$p_z$, and 4$s$ of Si atom) of the (SiH₂)₂ fragment (pull effect). This is unlike the end-on complex H₂SiNN[42,43]

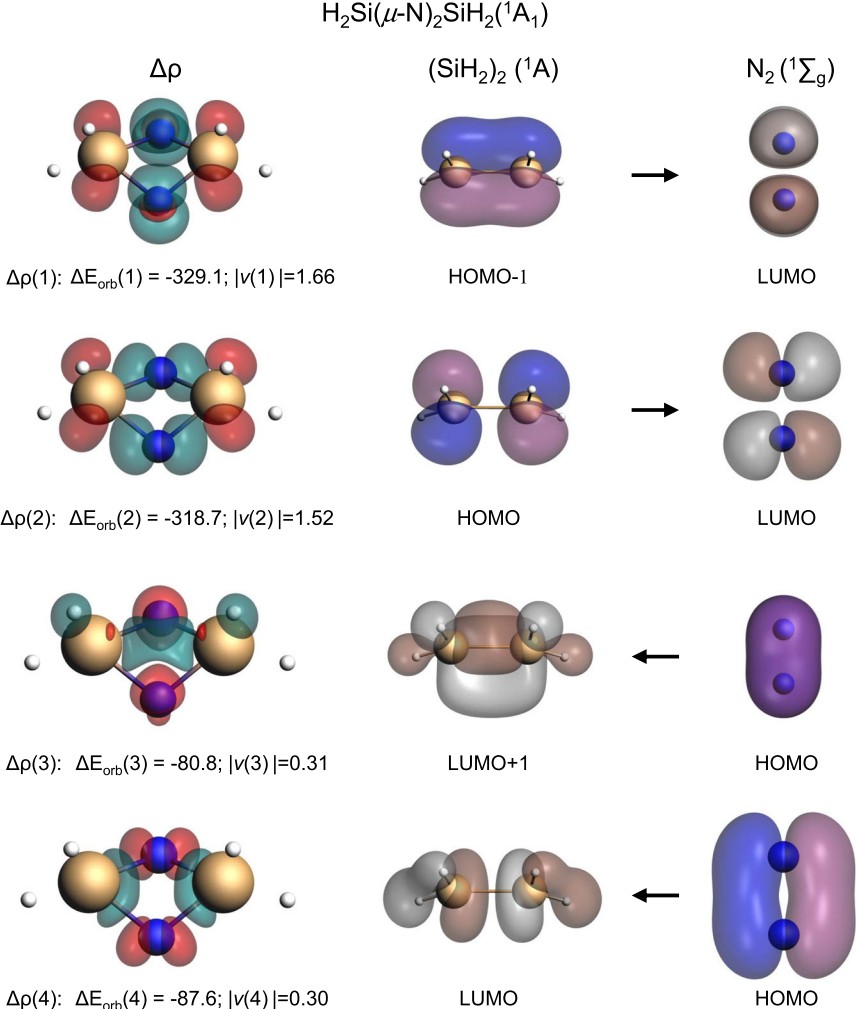

**Fig. 7 | Shape of the deformation densities, $\Delta\rho_{(1)\text{-}(4)}$ of $H_2Si(\mu\text{-N})_2SiH_2$ corresponding to $\Delta E_{orb(1)}\text{-}\Delta E_{orb(4)}$ and the associated fragment orbitals at the meta-Hybrid/ M06-2X/TZP level.** Isosurface values are 0.004 a.u. The eigenvalues $|v_n|$ give the size of the charge migration in e. The direction of the charge flow of the deformation densities is red→green.

with a σ-donation and a π back-donation, in which the σ-donation only plays a bonding role for the Si-N bond. However, for complex **A**, $H_2Si$: shows both interactions, which donates electron density to π*-antibonding orbitals of $N_2$ and removes electron density from the π-bonding orbitals of $N_2$ and, therefore, cooperatively cleaves the NN triple bond[69]. In addition, our EDA-NOCV results are illustrated in Supplementary Table 14 for **A** using neutral and charged fragments as interacting moieties. The smallest $\Delta E_{orb}$ values are found when the doubly charged species $(SiH_2)_2^{2+}$ ($^3A$) and $(N_2)^{2-}$ ($^3\Sigma_g$) are used for the calculations, which is a measure for the best description of the bonds finally formed[70,71]. The orbital term $\Delta E_{orb}$ accounts for 70% of the total attraction between the neutral units. However, the dominance of covalent bonding disappears when the final bonding situation is analyzed. The electrostatic part of the attractive interactions constitutes greater than 50% of the total attraction. The shape of the deformation densities, $\Delta\rho_{(1)\text{-}(4)}$ of $H_2Si(\mu\text{-N})_2SiH_2$ using $(SiH_2)_2^{2-}$ and $N_2^{2-}$ as interacting fragments were shown in Supplementary Fig. 27. When the bond finally formed, $(SiH_2)_2^{2+}$ ($^3B_2$) and $(N_2)^{2-}$ ($^3\Sigma_g$) still basically follow the rules of "push and pull", but the donation from $N_2^{2-}$ is stronger.

In this work, three activated dinitrogen species, $H_2Si(\mu\text{-N})_2SiH_2$, $H_2SiNN(H_2)$, and HNSiNH, have been identified by isotopic substitution experiments under cryogenic conditions in matrix-isolation experiments in conjunction with quantum-chemical calculations. The N≡N

triple bond was activated and broken by the synergistic interaction of two $SiH_2$ moieties in low-temperature matrix circumstances to form the stable aromatic ring system $Si_2N_2$. The SiNN is supposed to be an important starting compound to form the complex $H_2SiNN(H_2)$ (B) with $H_2$, which then further reacts with SiNN to build the final product $H_2Si(\mu\text{-N})_2SiH_2$ (A). The EDA-NOCV calculations support a dual interaction of $H_2Si$, which is able to donate electron density to π*-antibonding orbitals of $N_2$ and remove electron density from the π-bonding orbitals, leading to the cleavage of the $N_2$ triple bond and the formation of $H_2Si(\mu\text{-N})_2SiH_2$. This research work might open a different way to functionalize and activate dinitrogen molecules.

## Methods

### Matrix-isolation experiments

A Nd:YAG laser fundamental(1064 nm, 10 Hz repetition rate with 10 ns pulse width and 20−50 mJ/pulse) was focused on a rotating silicon target (Alfa Aesar), generating a bright plume. The laser-ablated silicon atoms reacting with $H_2$, $D_2$, HD, and $H_2 + D_2$ mixtures in solid $N_2$ and $^{15}N_2$, were condensed at 4 K using a closed-cycle helium refrigerator (Sumitomo Heavy Industries Model SRDK-408D2). Infrared spectra were recorded on a Bruker 80 v spectrometer at 0.5 cm$^{-1}$ resolution between 4000 and 400 cm$^{-1}$ using a HgCdTe range B detector. Further experimental details are provided in the Supplementary Information.

## Quantum chemistry calculation

All of the structures were optimized at Post-HF (CCSD(T)) and complementary density functional theory (DFT) methods and the vibration frequencies were computed analytically via Gaussian 09 program[72]. The bonding nature in **A** was investigated using energy decomposition analysis (EDA) combined with the natural orbitals for the chemical valence (NOCV) method. The analyses for the aromaticity were calculated with the Gaussian 09 program, the Multiwfn code[53], NBO 6.0 program[57], GIMIC2.0 program[47,73], and the RunEDDB program[49,50], respectively. Further quantum-chemical details are provided in the Supplementary Information.

## Data availability

All data generated in this study are provided in the Article and Supplementary Information. The experiment data that support the findings of this study are available from the corresponding author upon request.

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

## Acknowledgements

We gratefully acknowledge financial support from the National Natural Science Foundation of China (No. 22273066). Special thanks to Dr. Beckers for the generous discussions and support. Thanks are due to Hu Jin, Wen Liu, and Yan Lu for assistance with the calculation for aromaticity.

## Author contributions

B.X. and X.F.W. designed and supervised the matrix-isolation experiments and revised the manuscript. L.C. performed the experiments and the quantum-chemical calculations and wrote the draft. S.R. performed the quantum-chemical calculations and revised the manuscript. J.C and F.C. perform experiments partly.

## Competing interests

The authors declare no competing interest.
