## [Peer Review File · Nature Communications]

N₂ Cleavage by Silylene and Formation of H₂Si(μ-N)₂SiH₂Reviewers' Comments:

Reviewer #1:

Remarks to the Author:

The author proposed a nitrogen activation model with silicon-H₂ distinct from carbene nitrogen fixation. Initially, Si(0) coordinates with nitrogen, followed by oxidative addition of dihydrogen to form a H₂Si:N₂ species. Subsequently, a series of nitrogen activation processes ensue. This is highly intriguing, and I have no objections to these findings. It is foreseeable that achieving highly reactive free Si(0) species in synthetic chemistry is challenging. An open question arises: Is the H-substituted group in H₂Si:N₂ crucial for further enhancing nitrogen activation? This could have significant implications for nitrogen activation in synthetic chemistry.

Page2 line 34 Utilizing main group elements to achieve nitrogen activation may represent a novel reaction model with new chemical significance, beyond merely simulating d-block elements. I recommend that the author reconsider the conceptual framework of this sentence. Additionally, the review on main group nitrogen fixation (Chem. Soc. Rev. 2022, 51, 3846-3861.) should be cited to enhance readers' understanding of the development in this field.

Page2 line 37 "A substantially elongated N-N bond has been achieved" which should be described as a crucial metric for nitrogen activation, even though the stretching of nitrogen-nitrogen bonds is typically considered a key indicator of nitrogen activation.

Page2 line 41 While boron indeed serves as a classic example of nitrogen activation, carbenes, as elements of Group 14, have matured significantly in terms of nitrogen activation under matrix isolation conditions. Furthermore, carbenes appear to be more relevant to the topic. I recommend that the author cite relevant literature and provide a brief discussion.

Page2 line 52 The author describes the ability of silylenes to activate a range of small molecules, which indeed contributes to our understanding of their reactivity. Subsequently, nitrogen activation is also mentioned as a possibility. However, the logic in this sentence appears somewhat disjointed. In fact, silylenes do exhibit high reactivity, sharing similarities with certain carbenes that have a narrow HOMO-LUMO energy gap. Nevertheless, the challenge of activating more inert nitrogen by silylenes remains undiscussed. I recommend that the author briefly address the key difficulties in silylene-mediated nitrogen activation to enhance the significance of this work.

Page3 line 68 : The description 'silylenes: N₂ products' appears to be non-standard.

Page4 line 75 How does secondary irradiation (at 220 nm) specifically impact the generation of compound A? During A's formation, is it possible for it to proceed via interaction with an H₂SiN₂ side-on complex and another transient H₂Si: molecule, directly cleaving nitrogen, while B predominantly serves as the dissociator of N₂ in this process? Indeed, the nitrogen molecules in compound B have not been significantly activated. Furthermore, can the formation of H₂Si: be observed under an argon atmosphere without the use of nitrogen? If possible, could further introduction of nitrogen verify the presence of H₂Si: as another crucial reactive intermediate in the formation of compound A?

Page5 line 108: In Figure 1a, I noticed a broad peak around 1950 cm⁻¹, which may also indicate a N₂-complex. Could it be related to the formation of compound B, considering that this signal peak disappears in Figure 1b? Additionally, in Figure 1b, the signal intensity of NNSiNN sharply decreases, and under secondary irradiation (Figure 2e), the NNSiNN signal further diminishes. What is the relationship between NNSiNN, SiNN, H₂SiNN, and the generation of compounds B and A?"

Page5 line 147: How to explain the significant increase in SiH₄ signal intensity under 220 nm irradiation? I am curious whether there is a conversion relationship between compound B and SiH₄ during its formation process. If there is supporting evidence, please provide a brief description in the main text.

Page8 line 173: The N-N bond length is between 1.8-1.9 Å. It should be noted that there is a minor bonding interaction between two nitrogen atoms under this value. From the perspective of dinitrogen activation, in this case, the N-N bond is fully cleaved and each N part should be assigned as N³⁻. In the EDA-NOCV analysis, however, the authors have chosen two neutral fragments for comparison. The rationality of this choice should be discussed in the paper. The first two NOCV pairs cannot combine to form bonding orbitals, which is chemically meaningless.

To rule out the bonding nature, the fragmentation method with different charge and electron distribution should be tested by comparing their total value of the ΔE_{orb} term. After selecting reasonable interacting moieties, the bonding nature can be determined.

The author proposed a nitrogen activation model with silicon-H₂ distinct from carbene nitrogen fixation. Initially, Si(0) coordinates with nitrogen, followed by oxidative addition of dihydrogen to form a H₂Si:N₂ species. Subsequently, a series of nitrogen activation processes ensue. This is highly intriguing, and I have no objections to these findings. It is foreseeable that achieving highly reactive free Si(0) species in synthetic chemistry is challenging. An open question arises: Is the H-substituted group in H₂Si:N₂ crucial for further enhancing nitrogen activation? This could have significant implications for nitrogen activation in synthetic chemistry.

Reviewer #2:

Remarks to the Author:

Wang, Riedel and their co-workers reported a combined experimental and theoretical study on the dinitrogen activation by silylene. Three species including H₂Si(μ -N)₂SiH₂, H₂SiNN(H₂) and HNSiNH complexes were characterized by infrared spectroscopy in conjunction with quantum-chemical calculations. The topic is interesting and the findings of this manuscript is significant enough for the publication in this journal. Some issues are listed below for the authors' consideration.

1. The Lewis structure of compound A should be given. Currently, only a circle is drawn in Scheme 1. Its resonance structures should be provided and the dominated one should be specified if possible to help the readers understand the Si-N and N-N bonding because currently it is not clear whether the Si-N is a single or double bond.
2. Is the four-membered ring in species A aromatic or not even though it is puckered? The authors only provided the NICS values. Note that many aromaticity descriptors have been proposed on the basis of multiple manifestations of aromaticity. However, one must utilize a set instead of a single one to justify aromaticity in a given system (DOI: 10.1016/j.fmre.2023.04.004). The conclusion based on a single aromaticity index could be wrong. For instance, the recent results (DOI: 10.1021/jacs.3c07335) based on magnetic and electronic descriptors of aromaticity together with ¹¹B{¹H} NMR experimental spectra of boron-iodinated o-carboranes suggest that these two oxidized forms of a closo icosahedral dodecaiodo-dodecaborate cluster, [B₁₂I₁₂] and [B₁₂I₁₂]²⁺, behave as doubly 3D-aromatic compounds. However, an evaluation of the energetic contribution of the potential double 3D-aromaticity through homodesmotic reactions shows that delocalization in the I₁₂ shell, in contrast to the 10 σ -electron I₆²⁺ ring in the hexaiodobenzene dication, does not contribute to any stabilization of the system. Therefore, the [B₁₂I₁₂]^{0/2+} species cannot be considered as doubly 3D-aromatic. Thus, other indices e.g., EDDB or the energetic one should be used to confirm the findings in this manuscript.
3. Even for the NICS calculations, the authors could consider the CMO-NICS to separate the sigma and pi contributions. The last not the least, the NICS index is claimed to be a qualitative rather than quantitative method to evaluate aromaticity.
4. The recent study on din (DOI: 10.1002/jcc.27281; 10.1021/acs.inorgchem.1c03546; 10.1016/j.fmre.2023.04.004) should be discussed and cited.
5. Figure S8 should be moved from SI the manuscript and the Lewis structure of all the computed structures should be drawn.

Reviewer #3:

Remarks to the Author:

In this manuscript, the authors described the N₂ cleavage by the reaction of the parent silylene, H₂Si:, at low temperature in dinitrogen matrix resulting in the formation of H₂Si(m-N)₂ SiH₂. Small molecule activation is one of the hot topics in chemical transformation from the viewpoint of synthetic chemistry, and furthermore, the activation of nitrogen molecule with a new method is of particular importance in view of the development of effective supply of ammonia. The present work is very timely to show the intrinsic reactivity of the simplest silylene with nitrogen and the results here obtained will contribute further progress in the chemistry of metallylenes in small molecule activation. The experimental results are well interpreted in terms of the spectroscopic analyses in matrix and the identification of the reaction products is reasonably supported by the results of theoretical calculations.

Although the reviewer is essentially satisfied the results and conclusions of this work and recommends this manuscript for the publication in Nature Communications, the following points should be considered and adequate modification should be made before the final acceptance.

1) As for the reaction(s) examined in this manuscript as a subject, some schematic drawings should be added in the initial part of the text (not in the Supporting Information such as Figures S8 and S9). Otherwise, it might be somewhat difficult for the general readers unfamiliar with matrix isolation and main group chemistry. Especially, the silylene formation in the matrix with irradiation should be shown together with the subsequent reaction(s) with a nitrogen molecule.

2) The authors performed precise calculations for the reaction profile of the activation of a nitrogen molecule with H₂Si:. However, H₂Si: is only accessible in the low-temperature matrices. How about the case of more practical silylenes? Is it possible to show some information (the results of theoretical approach) for the reactions of typical models of substituted silylenes such as Me₂Si: or Ph₂Si: with a nitrogen molecule?

3) In addition, the evaluation of aromaticity of some molecule is very sensitive. Is the evaluation of A using NICS(1)_{av} most suitable? From the structural determination, the authors concluded A is not a planar but puckered molecule. In such case, it should be more careful to use the NICS evaluation. If the molecule is planar, NICS(1)_{zz} is more reliable. How about other evaluation methods for the aromaticity of A?

Responses

Reviewer 1

Q1: The author proposed a nitrogen activation model with silicon-H₂ distinct from carbene nitrogen fixation. Initially, Si(0) coordinates with nitrogen, followed by oxidative addition of dihydrogen to form a H₂Si:N₂ species. Subsequently, a series of nitrogen activation processes ensue. This is highly intriguing, and I have no objections to these findings. It is foreseeable that achieving highly reactive free Si(0) species in synthetic chemistry is challenging. An open question arises: Is the H-substituted group in H₂Si:N₂ crucial for further enhancing nitrogen activation? This could have significant implications for nitrogen activation in synthetic chemistry.

Answer: The structures and zero-point energies of dinitrogen activation models of substituted silylenes R₂Si: (F₂Si:, Cl₂Si:, Br₂Si:, (CN)₂Si:, (CF₃)₂Si:, Ph₂Si:, H₂Si: and (CH₃)₂Si:), such as R₂SiNN and R₂Si(μ-N)₂SiR₂ (R=F, Cl, Br, CN, CF₃, Ph, H and CH₃) were calculated on the singlet potential energy surfaces (PESs) at B3LYP/6-311++g (3df, 3pd) level (Figures R1 and S29). For the electron-withdrawing substitutions (F, Cl, Br, CN and CF₃), both the formation of the R₂SiNN and R₂Si(μ-N)₂SiR₂ is thermodynamically less favorable. However, for H₂Si:, Ph₂Si: and (CH₃)₂Si:, although the formation of R₂SiNN is slightly exothermic by 6.2 kcal/mol (H₂SiNN), and endothermic by 1.60 (Ph₂SiNN) and 0.06 ((CH₃)₂SiNN) kcal/mol, respectively. R₂SiNN will further react with another R₂Si to give R₂Si(μ-N)₂SiR₂ which is exothermic by 46.8, 56.60 and 46.06 kcal/mol, much higher than the ΔE of substituted silylenes of halogens (F, Cl, Br), CN and CF₃, suggesting that a synergistic interaction of two SiR₂ moieties with N₂ and ligand R without electron-withdraw are extremely important factors for activation of inert NN triple bond.

We added “Similar exothermic reactions could occur when H-substituted groups, such as CH₃ and Ph, are applied.” in line 195.

Figure R1. Potential energy surface for the reaction of $\text{H}_2\text{Si} + \text{N}_2 + \text{SiH}_2 \rightarrow \text{A}$ computed at the B3LYP/6-311++G(3df, 3pd) level of theory. The unit of relative energy is kcal/mol.

R	$\Delta E(\text{R}_2\text{SiNN})^{[a]}$	$\Delta E(\text{R}_2\text{Si}(\mu\text{-N})_2\text{SiR}_2)^{[b]}$
F	-0.19	-5.36
Cl	-0.08	-5.46
Br	-0.03	-9.60
CN	-1.49	-24.90
CF_3	-4.85	-35.89
Ph	0.06	-46.00
H	-6.21	-46.80
CH_3	1.60	-55.00

Figure S29. Structures and zero-point energies of R_2Si ·, R_2SiNN and $\text{R}_2\text{Si}(\mu\text{-N})_2\text{SiR}_2$ (R=F, Cl, Br, CN, CF_3 , Ph, H and CH_3). Calculated at B3LYP/6-311++g (3df, 3pd) level. [a] $\Delta E(\text{R}_2\text{SiNN}) = E(\text{R}_2\text{SiNN}) - E(\text{R}_2\text{Si}) - E(\text{NN})$; [b] $\Delta E(\text{R}_2\text{Si}(\mu\text{-N})_2\text{SiR}_2) = E(\text{R}_2\text{Si}(\mu\text{-N})_2\text{SiR}_2) - 2 \cdot E(\text{R}_2\text{Si}) - E(\text{NN})$. E (kcal/mol) is the zero-point energy of the corresponding complex optimized at B3LYP/6-311++g (3df, 3pd) level.

Q2. Page2 line 34 Utilizing main group elements to achieve nitrogen activation may represent a novel reaction model with new chemical significance, beyond merely simulating d-block elements. I recommend that the author reconsider the conceptual framework of this sentence. Additionally, the review on main group nitrogen fixation (*Chem. Soc. Rev.* 2022, 51, 3846-3861.) should be cited to enhance readers' understanding of the development in this field.

Answer: Thank you for your valuable suggestion. It's actually that nitrogen activation by main group elements is not only via simulating d-block elements by p-block elements (B and C), but also via many other reaction models (e.g. Ca, Sr, Ba and Be). Thus, we cited the related literature (*Chem. Soc. Rev.* 2022, 51, 3846-3861.) about main group nitrogen fixation at page2 line 35, reorganized the sentence and added the reaction model of s-block elements with dinitrogen.

Q3. Page2 line 37 "A substantially elongated N–N bond has been achieved" which should be described as a crucial metric for nitrogen activation, even though the stretching of nitrogen–nitrogen bonds is typically considered a key indicator of nitrogen activation.

Answer: This sentence has been revised to "A crucial metric for nitrogen activation of substantially elongated N–N bond has been achieved" at Page2 line 40.

Q4. Page2 line 41 While boron indeed serves as a classic example of nitrogen activation, carbenes, as elements of Group 14, have matured significantly in terms of nitrogen activation under matrix isolation conditions. Furthermore, carbenes appear to be more relevant to the topic. I recommend that the author cite relevant literature and provide a brief discussion.

Answer: The relevant literature about carbenes has been cited and a brief discussion has been added just after the introduction of boron at Page2 line 45-49. "Carbene, another reactive intermediate, has also been used for N₂ activation and conversion.¹⁷⁻¹⁹ Maier *et al.* found that singlet $\sigma^0\pi^2$ carbene (2-diazo-2H-imidazole) would bind with dinitrogen in the matrix, demonstrating the potential for $\sigma^0\pi^2$ carbene to activate dinitrogen.²⁰ Furthermore N₂ activation by a carbene pair has been calculated and the N≡N triple bond was predicted to be elongated to N–N single bond (1.428 Å) under the synergistic effect of the two CH₂ moieties.²¹"

Q5. Page2 line 52 The author describes the ability of silylenes to activate a range of small molecules, which indeed contributes to our understanding of their reactivity. Subsequently, nitrogen activation is also mentioned as a possibility. However, the logic in this sentence appears somewhat disjointed. In fact, silylenes do exhibit high reactivity, sharing similarities with certain carbenes that have a narrow HOMO-LUMO energy gap. Nevertheless, the challenge of activating more inert nitrogen by silylenes remains undiscussed. I recommend that the author briefly address the key difficulties in silylene-mediated nitrogen activation to enhance the significance of this work.

Answer: We added "The key difficulty in silylene-mediated nitrogen activation is to modify the occupied and vacant orbitals of silylene in space and energy, which could enhance weakening and functionalization of an inert chemical bond. For example, Driess *et al.* reported that two silylene moieties (bis-silylenes) could be cooperative to cleaving unreactive bonds, in which the Si---Si distance plays a crucial role.^{31,35}"

Q6. Page3 line 68: The description ‘silylenes: N2 products’ appears to be non-standard.

Answer: The description ‘silylenes: N2 products’ has been changed to ‘adduct products’ at Page 4 line 76.

Q7. Page 4 line 73 How does secondary irradiation (at 220 nm) specifically impact the generation of compound A? During A’s formation, is it possible for it to proceed via interaction with an H₂SiN₂ side-on complex and another transient H₂Si: molecule, directly cleaving nitrogen, while B predominantly serves as the dissociator of N₂ in this process? Indeed, the nitrogen molecules in compound B have not been significantly activated. Furthermore, can the formation of H₂Si: be observed under an argon atmosphere without the use of nitrogen? If possible, could further introduction of nitrogen verify the presence of H₂Si: as another crucial reactive intermediate in the formation of compound A?

Answer: Yes, the H₂Si: can be generated in argon matrix for sure (*J. Phys. Chem. A* **2002**, 106, 7696-7702.), but in nitrogen matrix the H₂Si: was stabilized as H₂SiNN because of nitrogen circumstance.

Compound A was observed upon the first $\lambda > 220$ nm irradiation. The following annealing had no impact to product **A** and **B**, but the precursor, but SiNN, can remove and diffuse in matrix and some of SiNN will be ready to interact with **B**. Further, the second $\lambda > 220$ nm irradiation will induce the reaction and give more compound **A**.

There is no evidence for **A**’s formation via interaction of an H₂SiN₂ side-on complex with another transient H₂Si:. First of all, H₂SiN₂ side-on complex is 21.9 kcal/mol higher in energy than that of end-on complex with B3LYP/6-311++g(3df,3pd). Secondly the significant NN stretching frequency is calculated at 1568.2(157) cm⁻¹, which was not observed in our experiments. Thirdly when H₂Si: is generated, which is immediately interacted with N₂ to form H₂SiNN in N₂ circumstance. What’s more, we carried out the Tesla coil discharge reactions of SiH₄ with or without H₂ in excess solid N₂, both H₂SiN₂ and species **B** can be observed while species **A** is missing due to the lack of SiNN (Figure S22 g-i).

It is actually that the formation of H₂Si: can be observed under an argon atmosphere without the use of nitrogen. {*J. Phys. Chem. A* **2002**, 106, 7696-7702.} However, when we tried to introduce to the N₂, there was no evidence for the formation of compound **A** (Figure S23) and the absorptions of H₂Si: showed no significant changes after irradiation.

Q8. Page5 line 108: In Figure 1a, I noticed a broad peak around 1950 cm⁻¹, which may also indicate a N₂-complex. Could it be related to the formation of compound B, considering that this signal peak disappears in Figure 1b? Additionally, in Figure 1b, the signal intensity of NNSiNN sharply decreases, and under secondary irradiation (Figure 2e), the NNSiNN signal further diminishes. What is the relationship between NNSiNN, SiNN, H₂SiNN, and the generation of compounds B and A?”

Answer: The broad band centered at 1950 cm⁻¹ is the baseline without correction. In the isotopic experiments, the infrared spectra of the laser-ablated Si atoms reactions with 10% D₂ in ¹⁴N₂ matrix and with 10% H₂ in ¹⁵N₂ matrix, neither ¹⁵N counterpart around 1885 cm⁻¹ (14N/15N isotopic ratio 1.034) in the laser-ablated Si atoms reactions with 10% H₂ in ¹⁵N₂ matrix nor D

counterpart around 1393 cm^{-1} (H/D isotopic ratio 1.400) in the laser-ablated Si atoms reactions with 10% D₂ in ¹⁴N₂ matrix was found. Thus, the broad band could be the uneven baseline rather than NN nor Si-H stretching modes.

The reaction between SiNN (³Σ⁻) and dinitrogen to give NNSiNN (¹A¹) is endothermic by 5.4 kcal/mol calculated with B3LYP/6-311++g(3df,3pd) and NNSiNN will diminish upon irradiation of the matrix with $\lambda = 313\text{ nm}$ (*Organometallics* **2000**, 19, 4775-4783.). To verify the relationship of NNSiNN and SiNN, laser-ablated Si atoms reactions with pure N₂ were done (Figure S18). The absorptions of NNSiNN diminished again and the absorptions of SiNN increased by 10% after $\lambda > 300\text{ nm}$ irradiation. In our experiments of laser-ablated silicon atoms with H₂ in dinitrogen matrix, SiNN (³Σ⁻) was supposed to be a starting compound to give the complex H₂SiNN (¹A₁) and complex **B** with H₂ upon $\lambda > 300\text{ nm}$ irradiation. It is certainly difficult to find H₂Si in solid nitrogen, which will react with N₂ in our experiment with no energy barrier. Thus, main product SiNN on codeposition was supposed to be the reactant to give product **A** with **B**.

Figure S18 Infrared spectra of the laser-ablated Si atoms reactions with pure ¹⁴N₂ at 4 K. (a) codeposition of Si + ¹⁴N₂ for 120 min; (b) after $\lambda > 300\text{ nm}$ irradiation for 10 min; (c) after $\lambda > 220\text{ nm}$ irradiation for 10 min.

Q9. Page5 line 147: How to explain the significant increase in SiH₄ signal intensity under 220 nm irradiation? I am curious whether there is a conversion relationship between compound B and SiH₄ during its formation process.

Answer: The fate of complex **B** either reacts with SiNN to form product **A**, or gives SiH₄ through H₂ bond broken under 220 nm irradiation. The complex **B** can be trapped because of N₂ matrix, otherwise the SiH₂ can react with H₂ to give SiH₄ with very small energy barrier of 1.4 kcal/mol (*Can. J. Chem.* **2000**, 78, 1428-1433).

Q10. Page8 line 173: The N-N bond length is between 1.8-1.9 Å. It should be noted that there is a minor bonding interaction between two nitrogen atoms under this value. From the perspective of dinitrogen activation, in this case, the N-N bond is fully cleaved and each N part should be assigned as N^3- . In the EDA-NOCV analysis, however, the authors have chosen two neutral fragments for comparison. The rationality of this choice should be discussed in the paper. The first two NOCV pairs cannot combine to form bonding orbitals, which is chemically meaningless. To rule out the bonding nature, the fragmentation method with different charge and electron distribution should be tested by comparing their total value of the ΔE_{orb} term. After selecting reasonable interacting moieties, the bonding nature can be determined.

Answer: The choice of the fragments is determined by the question of interest. Trivially, in a diatomic molecule X_2 the two atoms are chosen as interacting fragments, but when a heteroatomic species such as LiF shall be analyzed. The possibility of using different fragments for the bonding analysis provides more flexibility that may be used to address different questions about the bonding nature. Thus, the choice of neutral Li and F as interacting fragments includes all changes along the bond formation between the isolated atoms toward LiF, whereas the choice of the ions Li^+ and F^- addresses the question about the nature of the eventually formed bond. (*Chem. Rev.* **2019**, 119, 8781–8845).

In our work, the neutral fragments N_2 ($^1\Sigma_g$) and $(\text{SiH}_2)_2$ (^1A) in the singlet state, which refer to the symmetry-allowed dissociation products, have been selected as interacting moieties which addresses the question about all changes along the bond formation between two neutral fragments (page 9, line 197-199). We redo the EDA-NOCV calculation at the meta-Hybrid/M06-2X/TZP level instead of BP86/TZP level. The results are shown in Table S8. The first two NOCV pairs could combine to form bonding orbitals, which is chemically meaning now (Figure 4). The breakdown of the orbital interaction into pairwise orbital interactions reveals that the dominant orbital stabilization, $\Delta E_{\text{orb}(1)}$ ($-329.1 \text{ kcal mol}^{-1}$) and $\Delta E_{\text{orb}(2)}$ ($-318.7 \text{ kcal mol}^{-1}$), comes from the back-donation of the HOMO-1 (mainly $3p_x$ of the Si atom) and the HOMO (mainly $3p_x$ of the Si atom) of the $(\text{SiH}_2)_2$ moiety into the two perpendicular π^* MOs of the N_2 ligands, known as push effect. The most interesting orbital interactions are $\Delta E_{\text{orb}(3)}$ and $\Delta E_{\text{orb}(4)}$, which contribute to the donation of π MO electrons of the N_2 ligands to the LUMO+1 ($3p_z$ of Si atom and $1s$ of H atom) and LUMO (mainly $3s$, $3p_y$, $3p_z$ and $4s$ of Si atom) of the $(\text{SiH}_2)_2$ fragment (pull effect) (page 9, line 199-206).

Figure 4. Shape of the deformation densities, $\Delta\rho_{(1)-(4)}$ of $\text{H}_2\text{Si}(\mu\text{-N})_2\text{SiH}_2$ corresponding to $\Delta E_{\text{orb}(1)} - \Delta E_{\text{orb}(4)}$ and the associated fragment orbitals at the meta-Hybrid/M06-2X/TZP level. Isosurface values are 0.004 au. The eigenvalues $|v_n|$ give the size of the charge migration in e. The direction of the charge flow of the deformation densities is red→green.

Tables S13 and Figure S25 show the EDA-NOCV results for \mathbf{A} using neutral and charged fragments ($(\text{N}_2)^{2-}$, $(\text{N}_2)^4+$, and $(\text{N}_2)^{6-}$) as interacting moieties. The smallest ΔE_{orb} values are found when the doubly charged species $(\text{SiH}_2)_2^{2+}$ ($^3\text{B}_2$) and $(\text{N}_2)^{2-}$ ($^3\Sigma_g^-$) are used for the calculations, which is a measure for the best description of the bonds finally formed. The EDA-NOCV results in Table S13 suggest the orbital term ΔE_{orb} accounts for 70% of the total attraction between the neutral units, including the polarization within the fragments during bond formation. However, the dominance of covalent bonding disappears when the final bonding situation is analyzed. The electrostatic part of the attractive interactions constitutes greater than 50% of the total attraction. The shape of the deformation densities, $\Delta\rho_{(1)-(4)}$ of $\text{H}_2\text{Si}(\mu\text{-N})_2\text{SiH}_2$ using $(\text{SiH}_2)_2^{2+}$ and N_2^{2-} as interacting fragment were showing in Figure S25. When the bond finally formed, $(\text{SiH}_2)_2^{2+}$ ($^3\text{B}_2$) and $(\text{N}_2)^{2-}$ ($^3\Sigma_g^-$) still basically following the rules of "push and pull", but the back-donation from N_2^{2-} is stronger (page 9, line 210-219).

Table S13. EDA-NOCV results of $\text{H}_2\text{Si}(\mu\text{-N})_2\text{SiH}_2$ at the meta-Hybrid/M06-2X/TZP level taking $(\text{SiH}_2)_2$ and N_2 in the different charged states as interacting fragments. Energy values are in $\text{kcal}\cdot\text{mol}^{-1}$.

	(SiH ₂) ₂ (¹ A) + N ₂ (¹ Σ _g)	(SiH ₂) ₂ ²⁺ (³ A) + N ₂ ²⁻ (³ Σ _g)	(SiH ₂) ₂ ⁴⁺ (¹ A) + N ₂ ⁴⁺ (¹ Σ _g)	(SiH ₂) ₂ ⁶⁺ (¹ A) + N ₂ ⁶⁺ (¹ Σ _g)	
ΔE _{int}	-547.3	-1046.1	-2954.8	-6366.1	
ΔE _{Pauli}	811.4	792.3	840.0	912.1	
ΔE _{elstat} ^[a]	-405.2(29.8%)	-1142.2(62.1%)	-2937.1(77.4%)	-5492.6(75.5%)	
ΔE _{orb} ^[a]	-953.5(70.2%)	-696.3(37.9%)	-857.7(22.6%)	-1785.6(24.5%)	
ΔE ₁ ^[b]	-329.1(34.5%)	-110.3(15.8%)	-64.1(9.2%)	-230.7(26.9%)	-712.1(39.9%)
ΔE ₂ ^[b]	-318.7(33.4%)	-100.4(14.4%)	-58.0(8.3%)	-198.3(23.1%)	-311.0(17.4%)
ΔE ₃ ^[b]	-80.8(8.5%)	-78.6(11.3%)	-32.7(4.7%)	-107.6(12.5%)	-229.7(12.9%)
ΔE ₄ ^[b]	-87.5(9.2%)	-74.4(10.7%)	-32.4(4.6%)	-97.8(11.4%)	-116.2(6.5%)

Figure S25. Using (SiH₂)₂²⁺ and N₂²⁻ as interacting fragment and the shape of the deformation densities, Δρ₍₁₎₋₍₄₎ of H₂Si(μ-N)₂SiH₂ corresponding to ΔE_{orb(1)}-ΔE_{orb(4)} and the associated fragment orbitals at the meta-Hybrid/M06-2X/TZP level. Isosurface values are 0.004 au. The eigenvalues |v_n| give the size of the charge migration in e. The direction of the charge flow of the deformation densities is red→green.

Reviewer 2:

Wang, Riedel and their co-workers reported a combined experimental and theoretical study on the dinitrogen activation by silylene. Three species including H₂Si(μ-N)₂SiH₂, H₂SiNN(H₂) and HNSiNH complexes were characterized by infrared spectroscopy in conjunction with quantum-chemical calculations. The topic is interesting and the findings of this manuscript is significant enough for the publication in this journal. Some issues are listed below for the authors' consideration.

Q1. The Lewis structure of compound A should be given. Currently, only a circle is drawn in Scheme 1. Its resonance structures should be provided and the dominated one should be specified

if possible to help the readers understand the Si-N and N-N bonding because currently it is not clear whether the Si-N is a single or double bond.

Answer: The Lewis structure of compound A has been given in scheme 1. The resonance structures of compound A has been provided by NBO-based Natural Resonance Theory (NRT) analysis (Figure S26) at Page 5 Line 108-112. {*J. Comput. Chem.* **1998**, 19 (6), 593-609; *J. Comput. Chem.* **1998**, 19 (6), 610-627; *J. Comput. Chem.* **1998**, 19 (6), 628-646.} For the dominated one, Si-N is a single bond and two nitrogen atoms bond to give N-N single bond. However, our following atoms-in-molecules (AIM) and electron localization function (ELF) methodology find there is no covalent interaction between two nitrogen atoms but two aromatic π -electron delocalization in the cyclic compounds A with a molecular orbital (MO) analysis and computed canonical molecular orbital natural chemical shielding (CMO-NICS(1)_{zz}) values. (page 5, line 115-129)

Scheme 1. Binding modes of small molecules to silylene. (Dipp = 2,6-*i*-Pr₂C₆H₃; TBoN = N(SiMe₃){B(DippNCH)₂}; L = PhC(N*t*Bu)₂; X = 9, 9-dimethyl-xanthene- 4, 5-diyl or 1, 1' -ferrocenyl).

Figure S26. Natural Resonance Theory (NRT) description of product H₂Si(μ-N)₂SiH₂ (5

candidate reference structure).

Q2. Is the four-membered ring in species A aromatic or not even though it is puckered? The authors only provided the NICS values. Note that many aromaticity descriptors have been proposed on the basis of multiple manifestations of aromaticity. However, one must utilize a set instead of a single one to justify aromaticity in a given system (DOI: 10.1016/j.fmre.2023.04.004). The conclusion based on a single aromaticity index could be wrong. For instance, the recent results (DOI: 10.1021/jacs.3c07335) based on magnetic and electronic descriptors of aromaticity together with $^{11}\text{B}\{1\text{H}\}$ NMR experimental spectra of boron-iodinated o-carboranes suggest that these two oxidized forms of a closo icosahedral dodecaiodo-dodecaborate cluster, $[\text{B}_{12}\text{I}_{12}]$ and $[\text{B}_{12}\text{I}_{12}]^{2+}$, behave as doubly 3D-aromatic compounds. However, an evaluation of the energetic contribution of the potential double 3D-aromaticity through homodesmotic reactions shows that delocalization in the I12 shell, in contrast to the 10σ -electron I_6^{2+} ring in the hexaiodobenzene dication, does not contribute to any stabilization of the system. Therefore, the $[\text{B}_{12}\text{I}_{12}]^{0/2+}$ species cannot be considered as doubly 3D-aromatic. Thus, other indices e.g., EDDB or the energetic one should be used to confirm the findings in this manuscript.

Answer: Thank you for your valuable suggestion. The article (DOI: 10.1021/jacs.3c07335) evaluated aromaticity from three aspects: magnetic properties with the nucleus-independent chemical shift (NICS) method, electron delocalization properties with the electron density of delocalized bonds (EDDB_G) function and energy properties with aromatic stabilization energy (ASE) through homodesmotic reactions. In our article, more aspects except for magnetic properties should be taken into account.

Bond critical points (BCP) between both Si-N and N-N are found with atoms-in-molecules (AIM) {*Accounts. Chem. Res.* **1985**, 18, 9-15} methodology (Figure S13). The BCP between Si and N atoms locates in the region with positive Laplacian value and the accumulation of electronic charge, suggesting polar covalent interaction. However, a BCP between two nitrogen atoms may be found without a region of electronic-charge accumulation, which means the two atoms may not be bonded to each other. What's more, electron localization function (ELF) {*Angew. Chem. Int. Edit.* **1997**, 36, 1809-1832} for **A** also demonstrates that there is no covalent interaction between two N atoms (Figure S14). Thus, two aromatic π -electron delocalization in the cyclic compounds **A** was proposed following the $(4n+2)$ π electrons Hückel rule. For a molecular orbital (MO) analysis at B3LYP/6-311++G(3df,3pd) level with Gaussian 09, HOMO-6 orbital shows typical π -bonding orbitals (page 5, line 115-120) and the strongest π aromaticity is proved by the following CMO-NICS(1)_{zz} analysis. (page 6, line 126-129)

For electronic aspects, the multi-center bond order (MCBO), which is also known as multi-center index (MCI) {*Struct. Chem.* **1990**, 1, 423-427; *Phys. Chem. Chem. Phys.* **2016**, 18 (17), 11839-11846.} were calculated using the MultiWFN 3.8. The MCBO index of **A** is 0.3046, similar to that of $\text{FB}(\mu\text{-N})_2\text{BF}$ (0.3190) which also has an aromatic four-membered B_2N_2 ring {*Angew. Chem. Int. Edit.* **2021**, 60 (31), 17205-17210.} The analysis is added in the article at Page 5 Line 120-123.

Q3. Even for the NICS calculations, the authors could consider the CMO-NICS to separate the sigma and pi contributions. The last not the least, the NICS index is claimed to be a qualitative rather than quantitative method to evaluate aromaticity.

Answer: To separate the sigma and pi contributions of canonical molecular orbital, canonical molecular orbital natural chemical shielding (CMO-NICS(1)_{ZZ}) was calculated at B3LYP/ 6-311++G(3df,3pd) level with Gaussian 16 and NBO 6.0 program. From Figure S26, H₂Si(μ -N)₂SiH₂ (**A**) is found to be both the σ and π diatropic. It is the HOMO-2(σ) orbital making the largest diatropic contribution of -8.43 ppm for the σ aromaticity and HOMO-6(π) orbital making the largest diatropic contribution of -6.36 ppm for the π aromaticity. (page 6, line 126-129)

Q4. The recent study on din (DOI: 10.1002/jcc.27281; 10.1021/acs.inorgchem.1c03546; 10.1016/j.fmre.2023.04.004) should be discussed and cited.

Answer: The literature about dinitrogen activation by five-electron boron-centered radicals and five-electron boron-centered radicals (DOI: 10.1002/jcc.27281; 10.1021/acs.inorgchem.1c03546) has been discussed and cited at Page 2 Line 38-40. The review about the application of aromaticity and antiaromaticity (DOI: 10.1016/j.fmre.2023.04.004) has been discussed and cited at Page 5 Line 116.

Q5. Figure S8 should be moved from SI the manuscript and the Lewis structure of all the computed structures should be drawn.

Answer: Figure S8 has been moved from SI to the article (Figure 5) and the Lewis structures of products **A**, **B**, **C** were present in scheme 1.

Reviewer 3:

In this manuscript, the authors described the N₂ cleavage by the reaction of the parent silylene, H₂Si:, at low temperature in dinitrogen matrix resulting in the formation of H₂Si(μ -N)₂SiH₂. Small molecule activation is one of the hot topics in chemical transformation from the viewpoint of synthetic chemistry, and furthermore, the activation of nitrogen molecule with a new method is of particular importance in view of the development of effective supply of ammonia. The present work is very timely to show the intrinsic reactivity of the simplest silylene with nitrogen and the results here obtained will contribute further progress in the chemistry of metallylenes in small molecule activation. The experimental results are well interpreted in terms of the spectroscopic analyses in matrix and the identification of the reaction products is reasonably supported by the results of theoretical calculations. Although the reviewer is essentially satisfied the results and conclusions of this work and recommends this manuscript for the publication in Nature Communications, the following points should be considered and adequate modification should be made before the final acceptance.

Q1. As for the reaction(s) examined in this manuscript as a subject, some schematic drawings should be added in the initial part of the text (not in the Supporting Information such as Figures S8 and S9). Otherwise, it might be somewhat difficult for the general readers unfamiliar with matrix

isolation and main group chemistry. Especially, the silylene formation in the matrix with irradiation should be shown together with the subsequent reaction(s) with a nitrogen molecule.

Answer: Thank you. Figures S8 and S9 has been moved from SI to the article (Figures 5 - 6) to intuitively give a better understanding of the formation of product **A** and **B** (Figure 5), and the formation of product **C** (Figure 6).

Q2. The authors performed precise calculations for the reaction profile of the activation of a nitrogen molecule with H₂Si:. However, H₂Si: is only accessible in the low-temperature matrices. How about the case of more practical silylenes? Is it possible to show some information (the results of theoretical approach) for the reactions of typical models of substituted silylenes such as Me₂Si: or Ph₂Si: with a nitrogen molecule?

Answer: The structures and zero-point energies of dinitrogen activation models of substituted silylenes R₂Si: (F₂Si:, Cl₂Si:, Br₂Si:, (CN)₂Si:, (CF₃)₂Si:, Ph₂Si:, H and (CH₃)₂Si:), such as R₂SiNN and R₂Si(μ-N)₂SiR₂ (R=F, Cl, Br, CN, CF₃, Ph, H and CH₃) were calculated on the singlet potential energy surfaces (PESs) at B3LYP/6-311++g (3df, 3pd) level (Figures R1 and S29). For the electron-withdrawing substitutions (F, Cl, Br, CN and CF₃), both the formation of the R₂SiNN and R₂Si(μ-N)₂SiR₂ is thermodynamically less favorable. However, for H₂Si:, Ph₂Si: and (CH₃)₂Si:, although the formation of R₂SiNN is slightly exothermic by 6.2 kcal/mol (H₂SiNN) endothermic by 1.60 (Ph₂SiNN) and 0.06 ((CH₃)₂SiNN) kcal/mol, respectively, R₂SiNN will further react with another R₂Si to give R₂Si(μ-N)₂SiR₂ which is exothermic by 46.8, 56.60 and 46.06 kcal/mol, much higher than the ΔE of substituted silylenes of halogens (F, Cl, Br), CN and CF₃, suggesting a synergistic interaction of two SiR₂ moieties with N₂ and ligand R without electron-withdraw are extremely important factors for activation of inert NN triple bond.

R	$\Delta E(\text{R}_2\text{SiNN})^{[a]}$	$\Delta E(\text{R}_2\text{Si}(\mu\text{-N})_2\text{SiR}_2)^{[b]}$
F	-0.19	-5.36
Cl	-0.08	-5.46
Br	-0.03	-9.60
CN	-1.49	-24.90
CF ₃	-4.85	-35.89
Ph	0.06	-46.00
H	-6.21	-46.80
CH ₃	1.60	-55.00

Figure S29. Structures and zero-point energies of $\text{R}_2\text{Si}:$, R_2SiNN and $\text{R}_2\text{Si}(\mu\text{-N})_2\text{SiR}_2$ ($\text{R}=\text{F}$, Cl , Br , CN , CF_3 , Ph , H and CH_3). Calculated at B3LYP/6-311++g (3df, 3pd) level. [a] $\Delta E(\text{R}_2\text{SiNN}) = E(\text{R}_2\text{SiNN}) - E(\text{R}_2\text{Si}) - E(\text{NN})$; [b] $\Delta E(\text{R}_2\text{Si}(\mu\text{-N})_2\text{SiR}_2) = E(\text{R}_2\text{Si}(\mu\text{-N})_2\text{SiR}_2) - 2 \cdot E(\text{R}_2\text{Si}) - E(\text{NN})$. E (kcal/mol) is the zero-point energy of the corresponding complex optimized at B3LYP/6-311++g (3df, 3pd) level.

Q3. In addition, the evaluation of aromaticity of some molecule is very sensitive. Is the evaluation of A using NICS(1)av. most suitable? From the structural determination, the authors concluded A is not a planar but puckered molecule. In such case, it should be more careful to use the NICS evaluation. If the molecule is planar, NICS(1)zz is more reliable. How about other evaluation methods for the aromaticity of A ?

Answer: While the NICS index was originally obtained for planar aromatic systems, it has recently been suggested to calculate an average NICS(1)av index, $\text{NICS}(1)_{\text{av}} = [\text{NICS}(-1) + \text{NICS}(1)]/2$, as a probe of aromaticity in nonplanar molecular systems. {*RSC Adv.* **2016**, 6, 23900–23904}. We also calculated the anisotropy of the induced current density (AICD) but the result is not clear.

Except for the magnetic properties with the nucleus-independent chemical shift (NICS) method, we evaluate the aromaticity of **A** from mainly two aspects. First of all, two aromatic π -electron delocalization in the cyclic compounds **A** was proposed following the $(4n+2)$ π electrons Hückel rule. A BCP between two nitrogen atoms may be found without a region of electronic-charge accumulation in the atoms-in-molecules (AIM) {*Accounts. Chem. Res.* **1985**, 18, 9-15} methodology (Figure S13), which means the two atoms may not be bonded to each other. Electron localization function (ELF) {*Angew. Chem. Int. Edit.* **1997**, 36, 1809-1832} methodology find there is no covalent interaction between two nitrogen atoms (Figure S14) but two aromatic π -electron delocalization in the cyclic compounds **A** which is proved by a molecular orbital (MO) analysis (Figure S27) and computed canonical molecular orbital natural chemical shielding (CMO-NICS(1)zz) values (Figure S28). HOMO-6 orbital shows typical π -bonding orbitals (Figure S27) and makes the largest diatropic contribution of -6.36 ppm for the π aromaticity. (Figure S28)

What's more, the multi-center bond order (MCBO), which is also known as multi-center index (MCI) to evaluate aromaticity from the aspect of electron delocalization properties {*Struct. Chem.* **1990**, 1, 423-427; *Phys. Chem. Chem. Phys.* **2016**, 18 (17), 11839-11846.} were calculated using the MultiWFN 3.8. {*J. Comput. Chem.* **2012**, 33, 580–592} The MCBO index of **A** is 0.3046, similar to that of $\text{FB}(\mu\text{-N})_2\text{BF}$ (0.3190) which also has an aromatic four-membered B_2N_2 ring {*Angew. Chem. Int. Edit.* **2021**, 60 (31), 17205-17210.} The analysis is added in the article at Page 5 Line 115-129.

Figure S13. Laplacian distribution of the charge density of electron localization function (ELF) for $\text{H}_2\text{Si}(\mu\text{-N})_2\text{SiH}_2$ in the plane Si-N-N.

Figure S14. Isosurface map of electron localization function (ELF) for $\text{H}_2\text{Si}(\mu\text{-N})_2\text{SiH}_2$ in the plane Si-N-N.

HOMO-6

Figure S27. Selected frontier molecular orbitals of $\text{H}_2\text{Si}(\mu\text{-N})_2\text{SiH}_2$ (A) calculated at B3LYP /6-311++G(3df,3pd) level.

Figure S28. CMO-NICS(1)_{zz} of $\text{H}_2\text{Si}(\mu\text{-N})_2\text{SiH}_2$ (A) at B3LYP/6-311++G(3df,3pd) level. NICS values are in ppm. MO energies in a.u. are given on the left side.

Reviewers' Comments:

Reviewer #1:

Remarks to the Author:

After the careful revision, I would suggest the acceptance of this manuscript as it.

Reviewer #2:

Remarks to the Author:

Please check the attachment.

Wang, Riedel, Xu and their co-workers addressed most concerns in the first round of the review. Still, some issues should be solved before the publication of this manuscript.

1. The authors carried out additional NRT calculations to determine the Lewis structure of product $\text{H}_2\text{Si}(\mu\text{-N})_2\text{SiH}_2$ (Figure S26). Actually, it is quite reasonable to me. However, the authors mentioned that “However, our following atoms-in-molecules (AIM) and electron localization function (ELF) methodology find there is no covalent interaction between two nitrogen atoms but two aromatic π -electron delocalization in the cyclic compounds **A** with a molecular orbital (MO) analysis”. So how can the reader understand such a contradiction? Even the authors prefer an aromatic structure in the latter, the dominated Lewis structures (one or two) should be also provided to help the reader understand the structure. For instance, even for the aromatic benzene ring, two Lewis structures can be drawn.
2. The CMO-NICS calculations were also carried out. Again, it is problematic. As shown in Figure S28, three pi-MOs are present and the authors mentioned HOMO-6(π) orbital making the largest diatropic contribution of -6.36 ppm for the π aromaticity. (page 6, line 126-129). If that is the case, is it a two or six- π electrons in this species? How can two nitrogen atoms contribute two or six- π electrons?
3. The authors mentioned that the formation of R_2SiNN is slightly exothermic by 6.2 kcal/mol (H_2SiNN), indicating the unsubstituted one can produce the lowest reaction energy, which is similar to the previous finding on dinitrogen activation (DOI: 10.1016/j.ccl.2022.107759) and thus should be discussed and compared.
4. The page number of Ref.12 is not available. Thus, the DOI number rather than the page number should be used.
5. The computed activation energies in Figures 5 and 6 are both particularly high, is it in line with the experimental conditions?

Reviewer #3:

Remarks to the Author:

In the revised version, the authors made adequate modification according to the comments of reviewers. They added necessary information based on further experimental and theoretical results, which are appropriately appeared in the main text and supporting information. The reviewer is now satisfied with the responses to the referees' comments and the revised manuscript.

Responses

Reviewer 2:

Wang, Riedel, Xu and their co-workers addressed most concerns in the first round of the review. Still, some issues should be solved before the publication of this manuscript.

Q1. The authors carried out additional NRT calculations to determine the Lewis structure of product $\text{H}_2\text{Si}(\mu\text{-N})_2\text{SiH}_2$ (Figure S26). Actually, it is quite reasonable to me. However, the authors mentioned that “However, our following atoms-in-molecules (AIM) and electron localization function (ELF) methodology find there is no covalent interaction between two nitrogen atoms but two aromatic π -electron delocalization in the cyclic compounds **A** with a molecular orbital (MO) analysis”. So how can the reader understand such a contradiction? Even the authors prefer an aromatic structure in the latter, the dominated Lewis structures (one or two) should be also provided to help the reader understand the structure. For instance, even for the aromatic benzene ring, two Lewis structures can be drawn.

Answer: Thanks for your suggestion. The dominated Lewis structure calculated by NRT has been added in Scheme 1. To avoid the contradiction and based on the following analysis for the aromaticity, the contradictory words have been removed on Page 5 Lines 102.

Scheme 1. Binding modes of small molecules to silylene. (Dipp = 2,6- i -Pr₂C₆H₃; TBoN = N(SiMe₃){B(DippNCH)₂}; L = PhC(N*t*Bu)₂; X = 9, 9-dimethyl-xanthene- 4, 5-diyl or 1, 1' - ferrocenyl).

Q2. The CMO-NICS calculations were also carried out. Again, it is problematic. As shown in Figure S28, three pi-MOs are present and the authors mentioned HOMO-6(π) orbital making the largest diatropic contribution of -6.36 ppm for the π aromaticity. (page 6, lines 126-129). If that is the case, is it a two or six- π electrons in this species? How can two nitrogen atoms contribute two or six- π electrons?

Answer: To quantify delocalized electrons within the complex **A**, the electron density of delocalized bonds (EDDB) analysis was used and 1.55 electrons delocalization was calculated (Table S14) in which similar to $L_2Si_2P_2$ ($L = PhC(NtBu)_2$) where 1.77 electrons were found delocalization over the whole Si_2P_2 skeleton confirmed by the electron localizability index (ELI-d) and a natural bond orbital (NBO) analysis (*Angew. Chem. Int. Ed.* **50**, 2322-2325 (2011).) and the tetraatomic boron specie where 1.57 electrons were found delocalization and σ electrons dominated in the B_4 framework (*Organometallics* **39**, 2602-2608 (2020).). In our system, σ aromaticity is also the dominant one (60 %) in line with the CMO-NICS calculations. The EDDB analysis has been added on Page 5 Lines 108-109 and 120-124. 'Canonical molecular orbital natural chemical shielding (CMO-NICS(1)_{zz}) was calculated at B3LYP/ 6-311++G(3df,3pd) level to separate the σ and π contributions of canonical molecular orbital, and larger diatropic contribution of -14.6 ppm from σ orbitals compared with -11.9 ppm from π orbitals (Figure S28) indicates σ aromaticity dominated in **A**, consistent with the EDDB analysis where 1.55 electrons delocalization was calculated (Table S14) which are comparable to the number of delocalized electrons in $L_2Si_2P_2$ ($L = PhC(NtBu)_2$, 1.77 electrons) (*Angew. Chem. Int. Ed.* **50**, 2322-2325 (2011).) and the tetraatomic boron specie (1.57 electrons) (*Organometallics* **39**, 2602-2608 (2020).) and σ aromaticity is also the dominant one (60 %).'

What's more, the gauge including magnetically induced current (GIMIC) method is applied to obtain magnetically induced current densities in molecules and a net diamagnetic ring current can demonstrate the aromaticity (*J. Chem. Phys.* **121**, 3952-3963 (2004); *Phys. Chem. Chem. Phys.* **13**, 20500-20518 (2011).). The diatropic and paratropic ring currents for complex **A** are 29.6 nA T⁻¹ and -18.2 nA T⁻¹, respectively, yielding a net ring-current strength of 11.4 nA T⁻¹ calculated at B3LYP/6-311++g (3df, 3pd) level similar to the typical aromatic molecule benzene (11.8 nA T⁻¹) (*J. Phys. Chem. A* **113** (2009).) which means that complex **A** is an aromatic molecule (Figure S27). The GIMIC analysis has been added on Page 5 Lines 107-108 and 112-114. 'GIMIC method was calculated at B3LYP/6-311++g (3df, 3pd) level and a net diamagnetic ring current of 11.4 nA T⁻¹ in **A** similar to the typical aromatic molecule benzene (11.8 nA T⁻¹) (*J. Phys. Chem. A* **113**, 8668-8676 (2009)) could demonstrate the aromaticity of complex **A** (Figure S27).'

Table S14. Electron Density of Delocalized Bonds (EDDB) of $H_2Si(\mu-N)_2SiH_2$ (A**)**

complex	π/e	σ/e	Total/e
A	0.62	0.93	1.55

Figure S27. The iso-surface(left) and direction(right) of the ring-current density in $\text{H}_2\text{Si}(\mu\text{-N})_2\text{SiH}_2(\text{A})$ calculated at the B3LYP/6-311++g (3df, 3pd) level. Diatropic contribution is indicated with blue.

Q3. The authors mentioned that the formation of R_2SiNN is slightly exothermic by 6.2 kcal/mol (H_2SiNN), indicating the unsubstituted one can produce the lowest reaction energy, which is similar to the previous finding on dinitrogen activation (DOI: 10.1016/j.ccl.2022.107759) and thus should be discussed and compared.

Answer: The article (*Chinese. Chem. Lett.* **34**, 107759 (2023).) has been discussed and compared on Page 8 Lines 191-193. 'Similar to **NHC1-H**(*Chinese. Chem. Lett.* **34**, 107759 (2023).), H_2Si displays the smallest singlet–triplet energy gap and lowest ΔE to give H_2SiNN , showing the great potential for dinitrogen activation.' Except for $\Delta E(\text{R}_2\text{SiNN})$ and $\Delta E(\text{R}_2\text{Si}(\mu\text{-N})_2\text{SiR}_2)$, the singlet–triplet energy gap for complex R_2Si (ΔE_{ST}) is calculated to evaluate the reactivity of the reactants and the values are added in Figure S29. H_2Si displays the smallest singlet–triplet energy gap ($20.7 \text{ kcal mol}^{-1}$) and has the greatest potential for dinitrogen activation, compared with other substituted silylenes R_2Si : (F_2Si ·, Cl_2Si ·, Br_2Si ·, $(\text{CN})_2\text{Si}$ ·, $(\text{CF}_3)_2\text{Si}$ ·, Ph_2Si ·, and $(\text{CH}_3)_2\text{Si}$ ·). What's more, similar to **NHC1-H** which displays the lowest ΔG value, H_2Si also shows the lowest $\Delta E(\text{R}_2\text{SiNN})$.

Figure S29. Structures and zero-point energies of $R_2Si:$, R_2SiNN and $R_2Si(\mu-N)_2SiR_2$ ($R=F, Cl, Br, CN, CF_3, Ph, H$ and CH_3). Calculated at B3LYP/6-311++g (3df, 3pd) level.

R	$\Delta E(R_2SiNN)^{[a]}$	$\Delta E(R_2Si(\mu-N)_2SiR_2)^{[b]}$	$\Delta E_{ST}^{[c]}$
F	-0.19	-5.36	73.4
Cl	-0.08	-5.46	52.8
Br	-0.03	-9.60	47.8
CN	-1.49	-24.90	30.3
CF_3	-4.85	-35.89	27.9
Ph	0.06	-46.00	25.5
H	-6.21	-46.80	20.7
CH_3	1.60	-55.00	26.9

[a] $\Delta E(R_2SiNN) = E(R_2SiNN) - E(R_2Si) - E(NN)$; [b] $\Delta E(R_2Si(\mu-N)_2SiR_2) = E(R_2Si(\mu-N)_2SiR_2) - 2 \cdot E(R_2Si) - E(NN)$; [c] ΔE_{ST} is the singlet–triplet energy gap for complex $R_2Si:$; E (kcal/mol) is the zero-point energy of the corresponding complex optimized at B3LYP/6-311++g (3df, 3pd) level.

Q4. The page number of Ref.12 is not available. Thus, the DOI number rather than the page number should be used.

Answer: Thank you for pointing this out. The page number of Ref.12 is now updated and available on its website and Ref.12 has been changed to ‘Zeng, J., You, F. Y. & Zhu, J. Screening

seven-electron boron-centered radicals for dinitrogen activation. *J. Comput. Chem.* **45**, 648-654 (2024).’ at Page 11 line 254-255.

Q5. The computed activation energies in Figures 5 and 6 are both particularly high, is it in line with the experimental conditions?

Answer: The computed activation energies are actually in line with the experimental conditions since the reaction barrier can be overcome by the energy provided by photolysis and laser ablation in our experiments. In practice, there are at least five ways for atoms to be activated and thereby induced to react in matrix-isolation experiments, namely, by (i) photolysis, (ii) gaseous discharge, (iii) laser ablation, (iv) the chemical reaction giving rise to them, and (v) thermal means. (*Chem. Rev.* **102**, 4191-4241 (2002); *Russ. Chem. Rev.* **90**, 1142-1165 (2021).). Thus, in our experiments, both highly energetic laser-ablated silicon atoms and broad-band UV light ranging from 300 to 220 nm can provide energy to overcome the energy barrier. Photolysis is the most effective and controlled way to induce an atomic reaction and the nature of the excitation can be appraised with a well-defined wavelength. For example, neglecting energy loss, the energy provided by a Hg arc that emits broad-band UV-vis light (> 220 nm) is 5.6 eV (130 kcal mol⁻¹). When considering the energy loss, an activation energy of 229 kJ mol⁻¹ (55 kcal mol⁻¹) is also well within the reach of near-UV radiation (240 $<\lambda<$ 380 nm). (*J. Phys. Chem. A* **115**, 8638-8642 (2011).)

Upon $\lambda > 300$ nm irradiation, the complex H₂SiNN was observed (Figure 1b) with the activation energy (from SiNN and H₂) much less than 19.3 kcal mol⁻¹(Figure 5). Afterward, H₂SiNN reacts with a second H₂ molecule to form complex **B**. The corresponding barrier was computed to be only 1.8 kcal mol⁻¹ at DFT level which is in line with the increase of H₂SiNN and complex **B** and the decrease of SiNN upon $\lambda > 300$ nm irradiation in the experiments.

What’s more, complex **A** observed upon $\lambda > 220$ nm irradiation with the decrease of complex **B** and SiNN is in line with the highest activation barrier computed to be 38.4 kcal mol⁻¹. Although the activation barrier of Complex **C** is computed to be 32.8 kcal mol⁻¹, **C** is observed in freshly deposited samples and it increased markedly after $\lambda > 220$ nm irradiation, in which the laser-ablated energy provided in the codeposition process and the irradiation energy at $\lambda > 220$ nm most likely support the formation separately. Similarly, the barrier from SPN to cyc-PSN calculated to be 138 kJ mol⁻¹ (33 kcal mol⁻¹) can be overcome by UV light irradiation ($\lambda=255$ nm) (*Angew. Chem. Int. Ed.* **51**, 3334-3339 (2012).).

Reviewers' Comments:

Reviewer #2:

Remarks to the Author:

Wang, Riedel, Xu and their co-workers addressed most concerns in the second round of the review. Now, only one issue should be solved before the publication of this manuscript. That is the contradiction between the aromaticity and the Lewis structure. The authors show in Scheme 1 two lone pairs on the nitrogen atoms in the dominate resonance. Even all these lone pairs can be used to contribute to the aromaticity, it is only 4π electrons. How can a species with 4π electrons be aromatic? In addition, as the authors show a N-N single bond of A in Scheme 1. Is such a single bond supported by the bond order calculations? What are the Wiberg bond index or Fussy bond order of such a N-N bond in A?

Responses

Reviewer 2:

Wang, Riedel, Xu and their co-workers addressed most concerns in the second round of the review. Now, only one issue should be solved before the publication of this manuscript. That is the contradiction between the aromaticity and the Lewis structure. The authors show in Scheme 1 two lone pairs on the nitrogen atoms in the dominate resonance. Even all these lone pairs can be used to contribute to the aromaticity, it is only 4pi electrons. How can a species with 4pi electrons be aromatic?

In addition, as the authors show a N-N single bond of **A** in Scheme 1. Is such a single bond supported by the bond order calculations? What are the Wiberg bond index or Fussy bond order of such a N-N bond in **A**?

Answer: The two lone pairs of the nitrogen atoms are localized and don't exhibit delocalization characteristics in this species. Thus, the four lone electrons on the nitrogen atoms can't contribute to the aromaticity. For clarity, two lone pairs on the nitrogen atoms in the dominate resonance have been removed in Scheme 1 on Page 21(now Figure 1).

The Mayer bond order between N-N atoms in **A** is 0.689 calculated at B3LYP/6-311G(3df,3pd) level. In addition, Fussy bond order of 0.785 and Wiberg bond order of 1.088 are also obtained. The analysis for bond order is added on Page 5 Lines 104-105. 'A Mayer bond order of the N-N bond for molecule **A** is 0.689 computed at the B3LYP/6-311G(3df,3pd) level.'

Figure 1. Binding modes of small molecules to silylene. (Dipp = 2,6- i -Pr₂C₆H₃; TBoN = N(SiMe₃){B(DippNCH)₂}; L = PhC(N*t*Bu)₂; X = 9, 9-dimethyl-xanthene- 4, 5-diyl or 1, 1'-ferrocenyl).